# PERCEPTUAL ADVERSARIAL ROBUSTNESS: DEFENSE AGAINST UNSEEN THREAT MODELS

**Cassidy Laidlaw**
University of Maryland
claidlaw@umd.edu

**Sahil Singla**
University of Maryland
ssingla@cs.umd.edu

**Soheil Feizi**
University of Maryland
sfeizi@cs.umd.edu

## ABSTRACT

A key challenge in adversarial robustness is the lack of a precise mathematical characterization of human perception, used in the *definition* of adversarial attacks that are imperceptible to human eyes. Most current attacks and defenses try to avoid this issue by considering restrictive adversarial threat models such as those bounded by $L_2$ or $L_\infty$ distance, spatial perturbations, etc. However, models that are robust against any of these restrictive threat models are still fragile against other threat models, i.e. they have poor generalization to unforeseen attacks. Moreover, even if a model is robust against the union of several restrictive threat models, it is still susceptible to other imperceptible adversarial examples that are not contained in any of the constituent threat models. To resolve these issues, we propose adversarial training against the set of all imperceptible adversarial examples. Since this set is intractable to compute without a human in the loop, we approximate it using deep neural networks. We call this threat model the *neural perceptual threat model* (NPTM); it includes adversarial examples with a bounded *neural perceptual distance* (a neural network-based approximation of the true perceptual distance) to natural images. Through an extensive perceptual study, we show that the neural perceptual distance correlates well with human judgements of perceptibility of adversarial examples, validating our threat model. Under the NPTM, we develop novel perceptual adversarial attacks and defenses. Because the NPTM is very broad, we find that Perceptual Adversarial Training (PAT) against a perceptual attack gives robustness against many other types of adversarial attacks. We test PAT on CIFAR-10 and ImageNet-100 against five diverse adversarial attacks: $L_2$, $L_\infty$, spatial, recoloring, and JPEG. We find that PAT achieves state-of-the-art robustness against the union of these five attacks—more than doubling the accuracy over the next best model—*without* training against any of them. That is, PAT generalizes well to unforeseen perturbation types. This is vital in sensitive applications where a particular threat model cannot be assumed, and to the best of our knowledge, PAT is the first adversarial training defense with this property.

## 1 INTRODUCTION

Many modern machine learning algorithms are susceptible to adversarial examples: carefully crafted inputs designed to fool models into giving incorrect outputs (Biggio et al., 2013; Szegedy et al., 2014; Kurakin et al., 2016a; Xie et al., 2017). Much research has focused on increasing classifiers' robustness against adversarial attacks (Goodfellow et al., 2015; Madry et al., 2018; Zhang et al., 2019a). However, existing adversarial defenses for image classifiers generally consider simple threat models. An adversarial threat model defines a set of perturbations that may be made to an image in order to produce an adversarial example. Common threat models include $L_2$ and $L_\infty$ threat models, which constrain adversarial examples to be close to the original image in $L_2$ or $L_\infty$ distances. Some work has proposed additional threat models which allow spatial perturbations (Engstrom et al., 2017; Wong et al., 2019; Xiao et al., 2018), recoloring (Hosseini and Poovendran, 2018; Laidlaw and Feizi, 2019; Bhattad et al., 2019), and other modifications (Song et al., 2018; Zeng et al., 2019) of an image.

There are multiple issues with these unrealistically constrained adversarial threat models. First, hardening against one threat model assumes that an adversary will only attempt attacks within that threat model. Although a classifier may be trained to be robust against $L_\infty$ attacks, for instance,

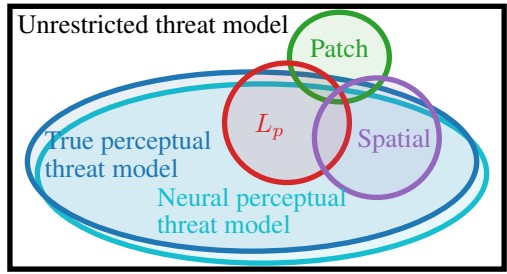

Figure 1: Relationships between various adversarial threat models. $L_p$ and spatial adversarial attacks are nearly contained within the perceptual threat model, while patch attacks may be perceptible and thus are not contained. In this paper, we propose a *neural perceptual threat model* (NPTM) that is based on an approximation of the true perceptual distance using neural networks.

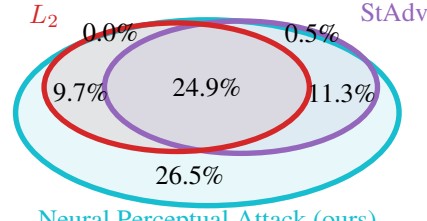

Figure 2: Area-proportional Venn diagram validating our threat model from Figure 1. Each ellipse indicates a set of vulnerable ImageNet-100 examples to one of three attacks: $L_2$, StAdv spatial (Xiao et al., 2018), and our neural perceptual attack (LPA, Section 4). Percentages indicate the proportion of test examples successfully attacked. Remarkably, the NPTM encompasses both other types of attacks and includes additional examples not vulnerable to either.

an attacker could easily generate a spatial attack to fool the classifier. One possible solution is to train against multiple threat models simultaneously (Jordan et al., 2019; Laidlaw and Feizi, 2019; Maini et al., 2019; Tramer and Boneh, 2019). However, this generally results in a lower robustness against any one of the threat models when compared to hardening against that threat model alone. Furthermore, not all possible threat models may be known at training time, and adversarial defenses do not usually generalize well to unforeseen threat models (Kang et al., 2019).

The ideal solution to these drawbacks would be a defense that is robust against a wide, unconstrained threat model. We differentiate between two such threat models. The *unrestricted adversarial threat model* (Brown et al., 2018) encompasses any adversarial example that is labeled as one class by a classifier but a different class by humans. On the other hand, we define the *perceptual adversarial threat model* as including all perturbations of natural images that are imperceptible to a human. Most existing narrow threat models such as $L_2$, $L_\infty$, etc. are near subsets of the perceptual threat model (Figure 1). Some other threat models, such as adversarial patch attacks (Brown et al., 2018), may perceptibly alter an image without changing its true class and as such are contained in the unrestricted adversarial threat model. In this work, we focus on the perceptual threat model.

The perceptual threat model can be formalized given the true perceptual distance $d^*(\mathbf{x}_1, \mathbf{x}_2)$ between images $\mathbf{x}_1$ and $\mathbf{x}_2$, defined as how different two images appear to humans. For some threshold $\epsilon^*$, which we call the perceptibility threshold, images $\mathbf{x}$ and $\mathbf{x}'$ are indistinguishable from one another as long as $d^*(\mathbf{x}, \mathbf{x}') \leq \epsilon^*$. Note that in general $\epsilon^*$ may depend on the specific input. Then, the perceptual threat model for a natural input $\mathbf{x}$ includes all adversarial examples $\widetilde{\mathbf{x}}$ which cause misclassification but are imperceptibly different from $\mathbf{x}$, i.e. $d^*(\mathbf{x}, \widetilde{\mathbf{x}}) \leq \epsilon^*$.

The true perceptual distance $d^*(\cdot, \cdot)$, however, cannot be easily computed or optimized against. To solve this issue, we propose to use a *neural perceptual distance*, an approximation of the true perceptual distance between images using neural networks. Fortunately, there have been many surrogate perceptual distances proposed in the computer vision literature such as SSIM (Wang et al., 2004). Recently, Zhang et al. (2018) discovered that comparing the internal activations of a convolutional neural network when two different images are passed through provides a measure, Learned Perceptual Image Patch Similarity (LPIPS), that correlates well with human perception. We propose to use the LPIPS distance $d(\cdot, \cdot)$ in place of the true perceptual distance $d^*(\cdot, \cdot)$ to formalize the *neural perceptual threat model* (NPTM).

We present adversarial attacks and defenses for the proposed NPTM. Generating adversarial examples bounded by the neural perceptual distance is difficult compared to generating $L_p$ adversarial examples because of the complexity and non-convexness of the constraint. However, we develop two attacks for the NPTM, **P**erceptual **P**rojected **G**radient **D**escent (**PPGD**) and **L**agrangian **P**erceptual **A**ttack (**LPA**) (see Section 4 for details). We find that LPA is by far the strongest adversarial attack at a given level of perceptibility (see Figure 4), reducing the most robust classifier studied to only 2.4%

accuracy on ImageNet-100 (a subset of ImageNet) while remaining imperceptible. LPA also finds adversarial examples outside of any of the other threat models studied (see Figure 2). Thus, even if a model is robust to many narrow threat models ($L_p$, spatial, etc.), LPA can still cause serious errors.

In addition to these attacks, which are suitable for evaluation of a classifier against the NPTM, we also develop Fast-LPA, a more efficient version of LPA that we use in **P**erceptual **A**dversarial **T**raining (**PAT**). Remarkably, using PAT to train a neural network classifier produces a single model with high robustness against a variety of imperceptible perturbations, including $L_\infty$, $L_2$, spatial, recoloring, and JPEG attacks, on CIFAR-10 and ImageNet-100 (Tables 2 and 3). For example, PAT on ImageNet-100 gives 32.5% accuracy against the *union* of these five attacks, whereas $L_\infty$ and $L_2$ adversarial training give 0.5% and 12.3% accuracy, respectively (Table 1). PAT achieves more than *double* the accuracy against this union of five threat models despite not explicitly training against any of them. Thus, it generalizes well to unseen threat models.

Does the LPIPS distance accurately reflect human perception when it is used to evaluate adversarial examples? We performed a study on Amazon Mechanical Turk (AMT) to determine how perceptible 7 different types of adversarial perturbations such as $L_\infty$, $L_2$, spatial, and recoloring attacks are at multiple threat-specific bounds. We find that LPIPS correlates well with human judgements across all the different adversarial perturbation types we examine. This indicates that the NPTM closely matches the true perceptual threat model and reinforces the utility of our perceptual attacks to measure adversarial robustness against an expansive threat model. Furthermore, this study allows calibration of a variety of attack bounds to a single perceptibility metric. We have released our dataset of adversarial examples along with the annotations made by participants for further study[1].

Table 1: ImageNet-100 accuracies against the union of five threat models ($L_\infty$, $L_2$, JPEG, StAdv, and Re-ColorAdv) for different adversarially trained (AT) classifiers and a single model trained using PAT.

| Training | Accuracy |
|----------|----------|
| PGD $L_\infty$ | 0.5% |
| PGD $L_2$ | 12.3% |
| **PAT (ours)** | **32.5%** |

## 2   RELATED WORK

**Adversarial robustness**     Adversarial robustness has been studied extensively for $L_2$ or $L_\infty$ threat models (Goodfellow et al., 2015; Carlini and Wagner, 2017; Madry et al., 2018) and non-$L_p$ threat models such as spatial perturbations (Engstrom et al., 2017; Xiao et al., 2018; Wong et al., 2019), recoloring of an image (Hosseini and Poovendran, 2018; Laidlaw and Feizi, 2019; Bhattad et al., 2019), and perturbations in the frequency domain (Kang et al., 2019). The most popular known adversarial defense for these threat models is adversarial training Kurakin et al. (2016b); Madry et al. (2018); Zhang et al. (2019a) where a neural network is trained to minimize the worst-case loss in a region around the input. Recent evaluation methodologies such as Unforeseen Attack Robustness (UAR) (Kang et al., 2019) and the Unrestricted Adversarial Examples challenge (Brown et al., 2018) have raised the problem of finding an adversarial defense which gives good robustness under more general threat models. Sharif et al. (2018) conduct a perceptual study showing that $L_p$ threat models are a poor approximation of the perceptual threat model. Dunn et al. (2020) and Xu et al. (2020) have developed adversarial attacks that manipulate higher-level, semantic features. Jin and Rinard (2020) train with a manifold regularization term, which gives some robustness to unseen perturbation types. Stutz et al. (2020) also propose a method which gives robustness against unseen perturbation types, but requires rejecting (abstaining on) some inputs.

**Perceptual similarity**     Two basic similarity measures for images are the $L_2$ distance and the Peak Signal-to-Noise Ratio (PSNR). However, these similarity measures disagree with human vision on perturbations such as blurring and spatial transformations, which has motivated others including SSIM (Wang et al., 2004), MS-SSIM (Wang et al., 2003), CW-SSIM (Sampat et al., 2009), HDR-VDP-2 (Mantiuk et al., 2011) and LPIPS (Zhang et al., 2018). MAD competition (Wang and Simoncelli, 2008) uses a constrained optimization technique related to our attacks to evaluate perceptual measures.

**Perceptual adversarial robustness**     Although LPIPS was previously proposed, it has mostly been used for development and evaluation of generative models (Huang et al., 2018; Karras et al., 2019). Jordan et al. (2019) first explored quantifying adversarial distortions with LPIPS distance. However, to the best of our knowledge, we are the first to apply a more accurate perceptual distance to the

---

[1]Code and data can be downloaded at `https://github.com/cassidylaidlaw/perceptual-advex`.

Original    Self-bd. LPA    External-bd. LPA    Original    Self-bd. LPA    External-bd. LPA

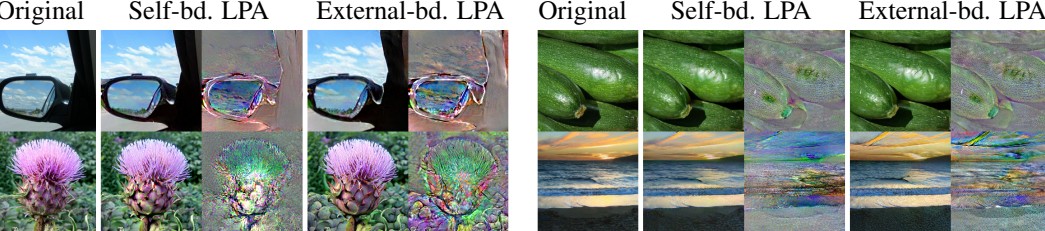

Figure 3: Adversarial examples generated using self-bounded and externally-bounded LPA perceptual adversarial attack (Section 4) with a large bound. Original images are shown in the left column and magnified differences from the original are shown to the right of the examples. See also Figure 7.

problem of improving adversarial robustness. As we show, adversarial defenses based on $L_2$ or $L_\infty$ attacks are unable to generalize to a more diverse threat model. Our method, PAT, is the first adversarial training method we know of that can generalize to unforeseen threat models without rejecting inputs.

## 3   NEURAL PERCEPTUAL THREAT MODEL (NPTM)

Since the true perceptual distance between images cannot be efficiently computed, we use approximations of it based on neural networks, i.e. neural perceptual distances. In this paper, we focus on the LPIPS distance (Zhang et al., 2018) while we note that other neural perceptual distances can also be used in our attacks and defenses.

Let $g : \mathcal{X} \to \mathcal{Y}$ be a convolutional image classifier network defined on images $\mathbf{x} \in \mathcal{X}$. Let $g$ have $L$ layers, and let the internal activations (outputs) of the $l$-th layer of $g(\mathbf{x})$ for an input $\mathbf{x}$ be denoted as $g_l(\mathbf{x})$. Zhang et al. (2018) have found that normalizing and then comparing the internal activations of convolutional neural networks correlates well with human similarity judgements. Thus, the first step in calculating the LPIPS distance using the network $g(\cdot)$ is to normalize the internal activations across the channel dimension such that the $L_2$ norm over channels at each pixel is one. Let $\hat{g}_l(\mathbf{x})$ denote these channel-normalized activations at the $l$-th layer of the network. Next, the activations are normalized again by layer size and flattened into a single vector $\phi(\mathbf{x}) \triangleq \left( \frac{\hat{g}_1(\mathbf{x})}{\sqrt{w_1 h_1}}, \ldots, \frac{\hat{g}_L(\mathbf{x})}{\sqrt{w_L h_L}} \right)$ where $w_l$ and $h_l$ are the width and height of the activations in layer $l$, respectively. The function $\phi : \mathcal{X} \to \mathcal{A}$ thus maps the inputs $\mathbf{x} \in \mathcal{X}$ of the classifier $g(\cdot)$ to the resulting normalized, flattened internal activations $\phi(\mathbf{x}) \in \mathcal{A}$, where $\mathcal{A} \subseteq \mathbb{R}^m$ refers to the space of all possible resulting activations. The LPIPS distance $d(\mathbf{x}_1, \mathbf{x}_2)$ between images $\mathbf{x}_1$ and $\mathbf{x}_2$ is then defined as:

$$d(\mathbf{x}_1, \mathbf{x}_2) \triangleq \|\phi(\mathbf{x}_1) - \phi(\mathbf{x}_2)\|_2 . \qquad (1)$$

In the original LPIPS implementation, Zhang et al. (2018) learn weights to apply to the normalized activations based on a dataset of human perceptual judgements. However, they find that LPIPS is a good surrogate for human vision even without the additional learned weights; this is the version we use since it avoids the need to collect such a dataset.

Now let $f : \mathcal{X} \to \mathcal{Y}$ be a classifier which maps inputs $\mathbf{x} \in \mathcal{X}$ to labels $f(\mathbf{x}) \in \mathcal{Y}$. $f(\cdot)$ could be the same as $g(\cdot)$, or it could be a different network; we experiment with both. For a given natural input $\mathbf{x}$ with the true label $y$, a neural perceptual adversarial example with a perceptibility bound $\epsilon$ is an input $\widetilde{\mathbf{x}} \in \mathcal{X}$ such that $\widetilde{\mathbf{x}}$ must be perceptually similar to $\mathbf{x}$ but cause $f$ to misclassify:

$$f(\widetilde{\mathbf{x}}) \neq y \qquad \text{and} \qquad d(\mathbf{x}, \widetilde{\mathbf{x}}) = \|\phi(\mathbf{x}) - \phi(\widetilde{\mathbf{x}})\|_2 \leq \epsilon. \qquad (2)$$

## 4   PERCEPTUAL ADVERSARIAL ATTACKS

We propose attack methods which attempt to find an adversarial example with small perceptual distortion. Developing adversarial attacks that utilize the proposed neural perceptual threat model is more difficult than that of standard $L_p$ threat models, because the LPIPS distance constraint is more complex than $L_p$ constraints. In general, we find an adversarial example that satisfies (2) by maximizing a loss function $\mathcal{L}$ within the LPIPS bound. The loss function we use is similar to the

margin loss from Carlini and Wagner (2017), defined as

$$\mathcal{L}(f(\mathbf{x}), y) \triangleq \max_{i \neq y} \big( z_i(\mathbf{x}) - z_y(\mathbf{x}) \big),$$

where $z_i(\mathbf{x})$ is the $i$-th logit output of the classifier $f(\cdot)$. This gives the constrained optimization

$$\max_{\widetilde{\mathbf{x}}} \quad \mathcal{L}(f(\widetilde{\mathbf{x}}), y) \qquad \text{subject to} \quad d(\mathbf{x}, \widetilde{\mathbf{x}}) = \|\phi(\mathbf{x}) - \phi(\widetilde{\mathbf{x}})\|_2 \leq \epsilon. \tag{3}$$

Note that in this attack problem, the classifier network $f(\cdot)$ and the LPIPS network $g(\cdot)$ are fixed. These two networks could be identical, in which case the same network that is being attacked is used to calculate the LPIPS distance that bounds the attack; we call this a *self-bounded* attack. If a different network is used to calculate the LPIPS bound, we call it an *externally-bounded* attack.

Based on this formulation, we propose two perceptual attack methods, Perceptual Projected Gradient Descent (PPGD) and Lagrangian Perceptual Attack (LPA). See Figures 3 and 7 for sample results.

**Perceptual Projected Gradient Descent (PPGD)**  The first of our two attacks is analogous to the PGD (Madry et al., 2018) attacks used for $L_p$ threat models. In general, these attacks consist of iteratively performing two steps on the current adversarial example candidate: (a) taking a step of a certain size under the given distance that maximizes a first-order approximation of the misclassification loss, and (b) projecting back onto the feasible set of the threat model.

Identifying the ideal first-order step is easy in $L_2$ and $L_\infty$ threat models; it is the gradient of the loss function and the sign of the gradient, respectively. However, computing this step is not straightforward with the LPIPS distance, because the distance metric itself is defined by a neural network. Following (3), we desire to find a step $\delta$ to maximize $\mathcal{L}(f(\mathbf{x}+\delta), y)$ such that $d(\mathbf{x}+\delta, \mathbf{x}) = \|\phi(\mathbf{x} + \delta) - \phi(\mathbf{x})\|_2 \leq \eta$, where $\eta$ is the step size. Let $\hat{f}(\mathbf{x}) := \mathcal{L}(f(\mathbf{x}), y)$ for an input $\mathbf{x} \in \mathcal{X}$. Let $J$ be the Jacobian of $\phi(\cdot)$ at $\mathbf{x}$ and $\nabla \hat{f}$ be the gradient of $\hat{f}(\cdot)$ at $\mathbf{x}$. Then, we can approximate (3) using a first-order Taylor's approximation of $\phi$ and $\hat{f}$ as follows:

$$\max_{\delta} \quad \hat{f}(\mathbf{x}) + (\nabla \hat{f})^\top \delta \qquad \text{subject to} \quad \|J\delta\|_2 \leq \eta. \tag{4}$$

We show that this constrained optimization can be solved in a closed form:

**Lemma 1.** *Let $J^+$ denote the pseudoinverse of $J$. Then the solution to (4) is given by*

$$\delta^* = \eta \frac{(J^\top J)^{-1}(\nabla \hat{f})}{\|(J^+)^\top (\nabla \hat{f})\|_2}.$$

See Appendix A.1 for the proof. This solution is still difficult to efficiently compute, since calculating $J^+$ and inverting $J^\top J$ are computationally expensive. Thus, we approximately solve for $\delta^*$ using the conjugate gradient method; see Appendix A.1 for details.

Perceptual PGD consists of repeatedly finding first-order optimal $\delta^*$ to add to the current adversarial example $\widetilde{\mathbf{x}}$ for a number of steps. Following each step, if the current adversarial example $\widetilde{\mathbf{x}}$ is outside the LPIPS bound, we project $\widetilde{\mathbf{x}}$ back onto the threat model such that $d(\widetilde{\mathbf{x}}, \mathbf{x}) \leq \epsilon$. The exact projection is again difficult due to the non-convexity of the feasible set. Thus, we solve it approximately with a technique based on Newton's method; see Algorithm 4 in the appendix.

**Lagrangian Perceptual Attack (LPA)**  The second of our two attacks uses a Lagrangian relaxation of the attack problem (3) similar to that used by Carlini and Wagner (2017) for constructing $L_2$ and $L_\infty$ adversarial examples. We call this attack the Lagrangian Perceptual Attack (LPA). To derive the attack, we use the following Lagrangian relaxation of (3):

$$\max_{\widetilde{\mathbf{x}}} \qquad \mathcal{L}(f(\widetilde{\mathbf{x}}), y) - \lambda \max \Big( 0, \|\phi(\widetilde{\mathbf{x}}) - \phi(\mathbf{x})\|_2 - \epsilon \Big). \tag{5}$$

The perceptual constraint cost, multiplied by $\lambda$ in (5), is designed to be 0 as long as the adversarial example is within the allowed perceptual distance; i.e. $d(\widetilde{\mathbf{x}}, \mathbf{x}) \leq \epsilon$; once $d(\widetilde{\mathbf{x}}, \mathbf{x}) > \epsilon$, however, it increases linearly by the LPIPS distance from the original input $\mathbf{x}$. Similar to $L_p$ attacks of Carlini and Wagner (2017), we adaptively change $\lambda$ to find an adversarial example within the allowed perceptual distance; see Appendix A.2 for details.

## 5 Perceptual Adversarial Training (PAT)

The developed perceptual attacks can be used to harden a classifier against a variety of adversarial attacks. The intuition, which we verify in Section 7, is that if a model is robust against neural

perceptual attacks, it can demonstrate an enhanced robustness against other types of unforeseen adversarial attacks. Inspired by adversarial training used to robustify models against $L_p$ attacks, we propose a method called Perceptual Adversarial Training (PAT).

Suppose we wish to train a classifier $f(\cdot)$ over a distribution of inputs and labels $(\mathbf{x}, y) \sim \mathcal{D}$ such that it is robust to the perceptual threat model with bound $\epsilon$. Let $\mathcal{L}_{\text{ce}}$ denote the cross entropy (negative log likelihood) loss and suppose the classifier $f(\cdot)$ is parameterized by $\theta_f$. Then, PAT consists of optimizing $f(\cdot)$ in a manner analogous to $L_p$ adversarial training (Madry et al., 2018):

$$\min_{\theta_f} \mathbb{E}_{(\mathbf{x},y)\sim\mathcal{D}} \left[ \max_{d(\widetilde{\mathbf{x}},\mathbf{x})\leq\epsilon} \mathcal{L}_{\text{ce}}(f(\widetilde{\mathbf{x}}), y) \right]. \tag{6}$$

The training formulation attempts to minimize the worst-case loss within a neighborhood of each training point $\mathbf{x}$. In PAT, the neighborhood is bounded by the LPIPS distance. Recall that the LPIPS distance is itself defined based on a particular neural network classifier. We refer to the normalized, flattened activations of the network used to define LPIPS as $\phi(\cdot)$ and $\theta_\phi$ to refer to its parameters. We explore two variants of PAT differentiated by the choice of the network used to define $\phi(\cdot)$. In *externally-bounded* PAT, a separate, pretrained network is used to calculate $\phi(\cdot)$, the LPIPS distance $d(\cdot, \cdot)$. In *self-bounded* PAT, the same network which is being trained for classification is used to calculate the LPIPS distance, i.e. $\theta_\phi \subseteq \theta_f$. Note that in self-bounded PAT the definition of the LPIPS distance changes during the training as the classifier is optimized.

The inner maximization in (6) is intractable to compute exactly. However, we can use the perceptual attacks developed in Section 4 to approximately solve it. Since the inner maximization must be solved repeatedly during the training process, we use an inexpensive variant of the LPA attack called Fast-LPA. In contrast to LPA, Fast-LPA does not search over values of $\lambda$. It also does not include a projection step at the end of the attack, which means it may sometimes produce adversarial examples outside the training bound. While this makes it unusable for evaluation, it is fine for training. Using Fast-LPA, PAT is nearly as fast as adversarial training; see Appendix A.3 and Algorithm 3 for details.

## 6 PERCEPTUAL EVALUATION

We conduct a thorough perceptual evaluation of our NPTM and attacks to ensure that the resulting adversarial examples are imperceptible. We also compare the perceptibility of perceptual adversarial attacks to five narrow threat models: $L_\infty$ and $L_2$ attacks, JPEG attacks (Kang et al., 2019), spatially transformed adversarial examples (StAdv) (Xiao et al., 2018), and functional adversarial attacks (ReColorAdv) (Laidlaw and Feizi, 2019). The comparison allows us to determine if the LPIPS distance is a good surrogate for human comparisons of similarity. It also allows us to set bounds across threat models with approximately the same level of perceptibility.

To determine how perceptible a particular threat model is at a particular bound (e.g. $L_\infty$ attacks at $\epsilon = 8/255$), we perform an experiment based on just noticeable differences (JND). We show pairs of images to participants on Amazon Mechanical Turk (AMT), an online crowdsourcing platform. In each pair, one image is a natural image from ImageNet-100 and one image is an adversarial perturbation of the natural image, generated using the particular attack against a classifier hardened to that attack. One of the images, chosen randomly, is shown for one second, followed by a blank screen for 250ms, followed by the second image for one second. Then, participants must choose whether they believe the images are the same or different. This procedure is identical to that used by Zhang et al. (2018) to originally validate the LPIPS distance. We report the proportion of pairs for which participants report the images are "different" as the *perceptibility* of the attack. In addition to adversarial example pairs, we also include sentinel image pairs which are exactly the same; only 4.1% of these were annotated as "different."

We collect about 1,000 annotations of image pairs for each of 3 bounds for all five threat models, plus our PPGD and LPA attacks (14k annotations total for 2.8k image pairs). The three bounds for each attack are labeled as small, medium, and large; bounds with the same label have similar perceptibility across threat models (see Appendix D Table 4). The dataset of image pairs and associated annotations is available for use by the community.

To determine if the LPIPS threat model is a good surrogate for the perceptual threat model, we use various classifiers to calculate the LPIPS distance $d(\cdot, \cdot)$ between the pairs of images used in the perceptual study. For each classifier, we determine the correlation between the mean LPIPS distance

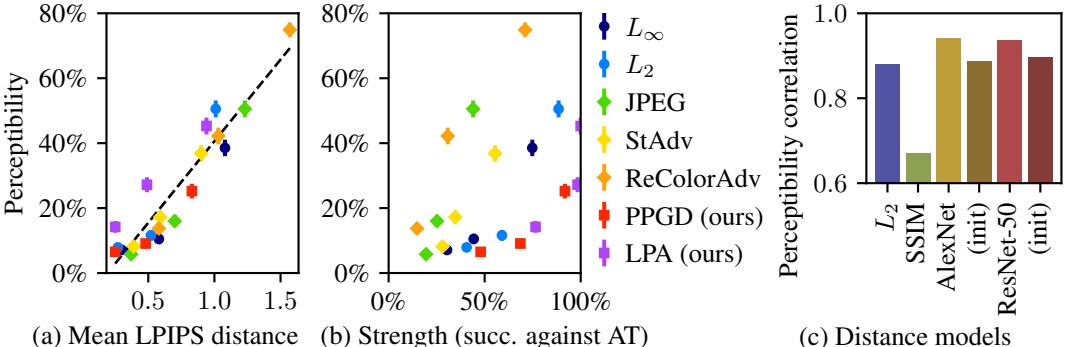

Figure 4: Results of the perceptual study described in Section 6 across five narrow threat models and our two perceptual attacks, each with three bounds. (a) The perceptibility of adversarial examples correlates well with the LPIPS distance (based on AlexNet) from the natural example. (b) The Lagrangian Perceptual Attack (LPA) and Perceptual PGD (PPGD) are strongest at a given perceptibility. Strength is the attack success rate against an adversarially trained classifier. (c) Correlation between the perceptibility of attacks and various distance measures: $L_2$, SSIM (Wang et al., 2004), and LPIPS (Zhang et al., 2018) calculated using various architectures, trained and at initialization.

it assigns to image pairs from each attack and the perceptibility of that attack (Figure 4c). We find that AlexNet (Krizhevsky et al., 2012), trained normally on ImageNet (Russakovsky et al., 2015), correlates best with human perception of these adversarial examples ($r = 0.94$); this agrees with Zhang et al. (2018) who also find that AlexNet-based LPIPS correlates best with human perception (Figure 4). A normally trained ResNet-50 correlates similarly, but not quite as well. Because AlexNet is the best proxy for human judgements of perceptual distance, we use it for all externally-bounded evaluation attacks. Note that even with an untrained network at initialization, the LPIPS distance correlates with human perception better than the $L_2$ distance. This means that even during the first few epochs of self-bounded PAT, the training adversarial examples are perceptually-aligned.

We use the results of the perceptual study to investigate which attacks are strongest at a particular level of perceptibility. We evaluate each attack on a classifier hardened against that attack via adversarial training, and plot the resulting success rate against the proportion of correct annotations from the perceptual study. Out of the narrow threat models, we find that $L_2$ attacks are the strongest for their perceptibility. However, our proposed PPGD and LPA attacks reduce a PAT-trained classifier to even lower accuracies (8.2% for PPGD and 0% for LPA), making it the strongest attack studied.

## 7 EXPERIMENTS

We compare Perceptual Adversarial Training (PAT) to adversarial training against narrow threat models ($L_p$, spatial, etc.) on CIFAR-10 (Krizhevsky and Hinton, 2009) and ImageNet-100 (the subset of ImageNet (Russakovsky et al., 2015) containing every tenth class by WordNet ID order). We find that PAT results in classifiers with robustness against a broad range of narrow threat models. We also show that our perceptual attacks, PPGD and LPA, are strong against adversarial training with narrow threat models. We evaluate with externally-bounded PPGD and LPA (Section 4), using AlexNet to determine the LPIPS bound because it correlates best with human judgements (Figure 4c). For $L_2$ and $L_\infty$ robustness evaluation we use AutoAttack (Croce and Hein, 2020), which combines four strong attacks, including two PGD variants and a black box attack, to give reliable evaluation.

**Evaluation metrics**  For both datasets, we evaluate classifiers' robustness to a range of threat models using two summary metrics. First, we compute the **union accuracy** against all narrow threat models ($L_\infty$, $L_2$, StAdv, ReColorAdv, and JPEG for ImageNet-100); this is the proportion of inputs for which a classifier is robust against *all* these attacks. Second, we compute the **unseen mean accuracy**, which is the mean of the accuracies against all the threat models not trained against; this measures how well robustness generalizes to other threat models.

**CIFAR-10**  We test ResNet-50s trained on the CIFAR-10 dataset with PAT and adversarial training (AT) against six attacks (see Table 2): $L_\infty$ and $L_2$ AutoAttack, StAdv (Xiao et al., 2018), ReCol-

Table 2: Accuracies against various attacks for models trained with adversarial training and Perceptual Adversarial Training (PAT) variants on CIFAR-10. Attack bounds are $8/255$ for $L_\infty$, 1 for $L_2$, 0.5 for PPGD/LPA (bounded with AlexNet), and the original bounds for StAdv/ReColorAdv. Manifold regularization is from Jin and Rinard (2020). See text for explanation of all terms.

| Training | Union | Unseen mean | Narrow threat models | | | | | NPTM | |
| --- | --- | --- | --- | --- | --- | --- | --- | --- | --- |
| | | | Clean | $L_\infty$ | $L_2$ | StAdv | ReColor | PPGD | LPA |
| Normal | 0.0 | 0.1 | 94.8 | 0.0 | 0.0 | 0.0 | 0.4 | 0.0 | 0.0 |
| AT $L_\infty$ | 1.0 | 19.6 | 86.8 | 49.0 | 19.2 | 4.8 | 54.5 | 1.6 | 0.0 |
| TRADES $L_\infty$ | 4.6 | 23.3 | 84.9 | 52.5 | 23.3 | 9.2 | 60.6 | 2.0 | 0.0 |
| AT $L_2$ | 4.0 | 25.3 | 85.0 | 39.5 | 47.8 | 7.8 | 53.5 | 6.3 | 0.3 |
| AT StAdv | 0.0 | 1.4 | 86.2 | 0.1 | 0.2 | 53.9 | 5.1 | 0.0 | 0.0 |
| AT ReColorAdv | 0.0 | 3.1 | 93.4 | 8.5 | 3.9 | 0.0 | 65.0 | 0.1 | 0.0 |
| AT all (random) | 0.7 | — | 85.2 | 22.0 | 23.4 | 1.2 | 46.9 | 1.8 | 0.1 |
| AT all (average) | 14.7 | — | 86.8 | 39.9 | 39.6 | 20.3 | 64.8 | 10.6 | 1.1 |
| AT all (maximum) | 21.4 | — | 84.0 | 25.7 | 30.5 | 40.0 | 63.8 | 8.6 | 1.1 |
| Manifold reg. | 21.2 | 36.2 | 72.1 | 36.8 | 43.4 | 28.4 | 63.1 | 8.7 | 9.1 |
| PAT-self | 21.9 | 45.6 | 82.4 | 30.2 | 34.9 | 46.4 | 71.0 | 13.1 | 2.1 |
| PAT-AlexNet | **27.8** | **48.5** | 71.6 | 28.7 | 33.3 | 64.5 | 67.5 | **26.6** | **9.8** |

Table 3: Comparison of adversarial training against narrow threat models and Perceptual Adversarial Training (PAT) on ImageNet-100. Accuracies are shown against seven attacks with the medium bounds from Table 4. PAT greatly improves accuracy (33% vs 12%) against the union of the narrow threat models despite not training against any of them. See text for explanation of all terms.

| Training | Union | Unseen mean | Narrow threat models | | | | | | NPTM | |
| --- | --- | --- | --- | --- | --- | --- | --- | --- | --- | --- |
| | | | Clean | $L_\infty$ | $L_2$ | JPEG | StAdv | ReColor | PPGD | LPA |
| Normal | 0.0 | 0.1 | 89.1 | 0.0 | 0.0 | 0.0 | 0.0 | 2.4 | 0.0 | 0.0 |
| $L_\infty$ | 0.5 | 11.3 | 81.7 | 55.7 | 3.7 | 10.8 | 4.6 | 37.5 | 1.5 | 0.0 |
| $L_2$ | 12.3 | 31.5 | 75.3 | 46.1 | 41.0 | 56.6 | 22.8 | 31.2 | 22.0 | 0.5 |
| JPEG | 0.1 | 7.4 | 84.8 | 13.7 | 1.8 | 74.8 | 0.3 | 21.0 | 0.5 | 0.0 |
| StAdv | 0.6 | 2.1 | 77.1 | 2.6 | 1.2 | 3.7 | 65.3 | 2.9 | 0.6 | 0.0 |
| ReColorAdv | 0.0 | 0.1 | 90.1 | 0.2 | 0.0 | 0.1 | 0.0 | 69.3 | 0.0 | 0.0 |
| All (random) | 0.9 | — | 78.6 | 38.3 | 26.4 | 61.3 | 1.4 | 32.5 | 16.1 | 0.2 |
| PAT-self | **32.5** | **46.4** | 72.6 | 45.0 | 37.7 | 53.0 | 51.3 | 45.1 | 29.2 | **2.4** |
| PAT-AlexNet | 25.5 | 44.7 | 75.7 | 46.8 | 41.0 | 55.9 | 39.0 | 40.8 | **31.1** | 1.6 |

orAdv (Laidlaw and Feizi, 2019), and PPGD and LPA. This allows us to determine if PAT gives robustness against a range of adversarial attacks. We experiment with using various models to calculate the LPIPS distance during PAT. We try using the same model both for classification and to calculate the LPIPS distance (self-bounded PAT). We also use AlexNet trained on CIFAR-10 prior to PAT (externally-bounded PAT). We find that PAT outperforms $L_p$ adversarial training and TRADES (Zhang et al., 2019a), improving the union accuracy from <5% to >20%, and nearly doubling mean accuracy against unseen threat models from 26% to 49%. Surprisingly, we find that PAT even outperforms threat-specific AT against StAdv and ReColorAdv; see Appendix F.5 for more details. Ablation studies of PAT are presented in Appendix F.1.

**ImageNet-100** We compare ResNet-50s trained on the ImageNet-100 dataset with PAT and adversarial training (Table 3). Classifiers are tested against seven attacks at the medium bound from the perceptual study (see Section 6 and Appendix Table 4). Self- and externally-bounded PAT give similar results. Both produce more than double the next highest union accuracy and also significantly increase the mean robustness against unseen threat models by around 15%.

**Perceptual attacks** On both CIFAR-10 and ImageNet-100, we find that Perceptual PGD (PPGD) and Lagrangian Perceptual Attack (LPA) are the strongest attacks studied. LPA is the strongest,

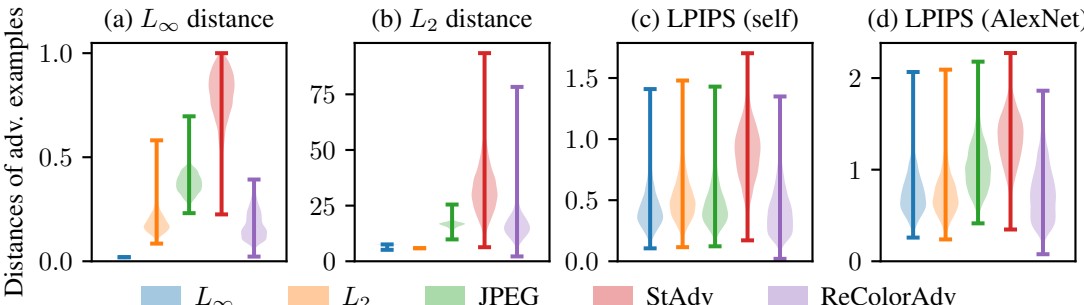

Figure 5: We generate samples via different adversarial attacks using narrow threat models in ImageNet-100 and measure their distances from natural inputs using $L_p$ and LPIPS metrics. The distribution of distances for each metric and threat model is shown as a violin plot. (a-b) $L_p$ metrics assign vastly different distances across perturbation types, making it impossible to train against all of them using $L_p$ adversarial training. (c-d) LPIPS assigns similar distances to similarly perceptible attacks, so a single training method, PAT, can give good robustness across different threat models.

reducing the most robust classifier to 9.8% accuracy on CIFAR-10 and 2.4% accuracy on ImageNet-100. Also, models most robust to LPA in both cases are those that have the best union and unseen mean accuracies. This demonstrates the utility of evaluating against LPA as a proxy for adversarial robustness against a range of threat models. See Appendix E for further attack experiments.

**Comparison to other defenses against multiple attacks**    Besides the baseline of adversarially training against a single attack, we also compare PAT to adversarially training against multiple attacks (Tramer and Boneh, 2019; Maini et al., 2019). We compare three methods for multiple-attack training: choosing a random attack at each training iteration, optimizing the average loss across all attacks, and optimizing the maximum loss across all attacks. The latter two methods are very expensive, increasing training time by a factor equal to the number of attacks trained against, so we only evaluate these methods on CIFAR-10. As in Tramer and Boneh (2019), we find that the maximum loss strategy leads to the greatest union accuracy among the multiple-attack training methods. However, PAT performs even better on CIFAR-10, *despite training against none of the attacks and taking one fourth of the time to train*. The random strategy, which is the only feasible one on ImageNet-100, performs much worse than PAT. Even the best multiple-attack training strategies still fail to generalize to the unseen neural perceptual attacks, PPGD and LPA, achieving much lower accuracy than PAT.

On CIFAR-10, we also compare PAT to manifold regularization (MR) (Jin and Rinard, 2020), a non-adversarial training defense. MR gives union accuracy close to PAT-self, but much lower clean accuracy; for PAT-AlexNet, which gives similar clean accuracy to MR, the union accuracy is much higher.

**Threat model overlap**    In Figure 2, we investigate how the sets of images vulnerable to $L_2$, spatial, and perceptual attacks overlap. Nearly all adversarial examples vulnerable to $L_2$ or spatial attacks are also vulnerable to LPA. However, there is only partial overlap between the examples vulnerable to $L_2$ and spatial attacks. This helps explain why PAT results in improved robustness against spatial attacks (and other diverse threat models) compared to $L_2$ adversarial training.

**Why does PAT work better than $L_p$ adversarial training?**    In Figure 5, we give further explanation of why PAT results in improved robustness against diverse threat models. We generate many adversarial examples for the $L_\infty$, $L_2$, JPEG, StAdv, and ReColorAdv threat models and measure their distance from the corresponding natural inputs using $L_p$ distances and the neural perceptual distance, LPIPS. While $L_p$ distances vary widely, LPIPS gives remarkably comparable distances to different types of adversarial examples. Covering all threat models during $L_\infty$ or $L_2$ adversarial training would require using a huge training bound, resulting in poor performance. In contrast, PAT can obtain robustness against all the narrow threat models at a reasonable training bound.

**Robustness against common corruptions**    In addition to evaluating PAT against adversarial examples, we also evaluate its robustness to random perturbations in the CIFAR-10-C and ImageNet-C datasets (Hendrycks and Dietterich, 2019). We find that PAT gives increased robustness (lower relative mCE) against these corruptions compared to adversarial training; see Appendix G for details.

## 8  CONCLUSION

We have presented attacks and defenses for the neural perceptual threat model (realized by the LPIPS distance) and shown that it closely approximates the true perceptual threat model, the set of all perturbations to natural inputs which fool a model but are imperceptible to humans. Our work provides a novel method for developing defenses against adversarial attacks that generalize to unforeseen threat models. Our proposed perceptual adversarial attacks and PAT could be extended to other vision algorithms, or even other domains such as audio and text.

## ACKNOWLEDGMENTS

This project was supported in part by NSF CAREER AWARD 1942230, HR 00111990077, HR00112090132, HR001119S0026, NIST 60NANB20D134, AWS Machine Learning Research Award and Simons Fellowship on "Foundations of Deep Learning."

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

# APPENDIX

## A  PERCEPTUAL ATTACK ALGORITHMS

### A.1  PERCEPTUAL PGD

Recall from Section 4 that Perceptual PGD (PPGD) consists of repeatedly applying two steps: a first-order step in LPIPS distance to maximize the loss, followed by a projection into the allowed set of inputs. Here, we focus on the first-order step; see Appendix A.4 for how we perform projection onto the LPIPS ball.

We wish to solve the following constrained optimization for the step $\delta$ given the step size $\eta$ and current input $\mathbf{x}$:

$$\max_{\delta} \quad \mathcal{L}(f(\mathbf{x} + \delta), y) \qquad \text{subject to} \quad \|J\delta\|_2 \leq \eta \tag{7}$$

Let $\hat{f}(\mathbf{x}) := \mathcal{L}(f(\mathbf{x}), y)$ for an input $\mathbf{x} \in \mathcal{X}$. Let $J$ be the Jacobian of $\phi(\cdot)$ at $\mathbf{x}$ and $\nabla\hat{f}$ be the gradient of $\hat{f}(\cdot)$ at $\mathbf{x}$.

**Lemma 1.** *The first-order approximation of (7) is*

$$\max_{\delta} \quad \hat{f}(\mathbf{x}) + (\nabla\hat{f})^\top \delta \qquad \text{subject to} \quad \|J\delta\|_2 \leq \eta, \tag{8}$$

*and can be solved in closed-form by*

$$\delta^* = \eta \frac{(J^\top J)^{-1}(\nabla\hat{f})}{\|(J^+)^\top(\nabla\hat{f})\|_2}.$$

*where $J^+$ is the pseudoinverse of $J$.*

*Proof.* We solve (8) using Lagrange multipliers. First, we take the gradient of the objective:

$$\nabla_\delta \left[ \hat{f}(\mathbf{x}) + (\nabla\hat{f})^\top \delta \right] = \nabla\hat{f}$$

We can rewrite the constraint by squaring both sides to obtain

$$\delta^\top J^\top J\delta - \epsilon^2 \leq 0$$

Taking the gradient of the constraint gives

$$\nabla_\delta \left[ \delta^\top J^\top J\delta - \epsilon^2 \right] = 2J^\top J\delta$$

Now, we set one gradient as a multiple of the other and solve for $\delta$:

$$J^\top J\delta = \lambda(\nabla\hat{f}) \tag{9}$$

$$\delta = \lambda(J^\top J)^{-1}(\nabla\hat{f}) \tag{10}$$

Substituting into the constraint from (8) gives

$$\|J\delta\|_2 = \eta$$
$$\|J\lambda(J^\top J)^{-1}(\nabla\hat{f})\|_2 = \eta$$
$$\lambda\|J(J^\top J)^{-1}(\nabla\hat{f})\|_2 = \eta$$
$$\lambda\|((J^\top J)^{-1}J^\top)^\top(\nabla\hat{f})\|_2 = \eta$$
$$\lambda\|(J^+)^\top(\nabla\hat{f})\|_2 = \eta$$
$$\lambda = \frac{\eta}{\|(J^+)^\top(\nabla\hat{f})\|_2}$$

We substitute this value of $\lambda$ into (10) to obtain

$$\delta^* = \eta \frac{(J^\top J)^{-1}(\nabla\hat{f})}{\|(J^+)^\top(\nabla\hat{f})\|_2}. \tag{11}$$

∎

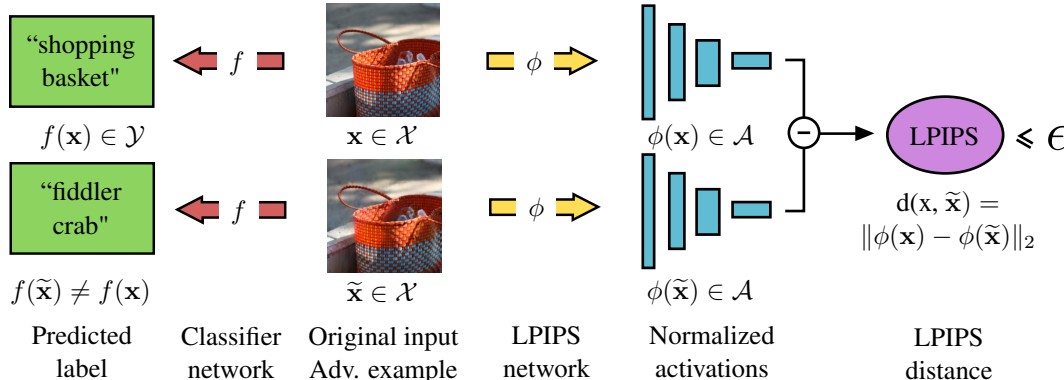

Figure 6: Creating an adversarial example in the LPIPS threat model.

**Solution with conjugate gradient method**    Calculating (11) directly is computationally intractable for most neural networks, since inverting $J^\top J$ and calculating the pseudoinverse of $J$ are expensive. Instead, we approximate $\delta^*$ by using the conjugate gradient method to solve the following linear system, based on (9):

$$J^\top J \delta = \nabla \hat{f} \qquad (12)$$

$\nabla \hat{f}$ is easy to calculate using backpropagation. The conjugate gradient method does not require calculating fully $J^\top J$; instead, it only requires the ability to perform matrix-vector products $J^\top J v$ for various vectors $v$.

We can approximate $Jv$ using finite differences given a small, positive value $h$:

$$Jv \approx \frac{\phi(\mathbf{x} + hv) - \phi(\mathbf{x})}{h}$$

Then, we can calculate $J^\top J v$ by introducing an additional variable $u$ and using autograd:

$$\nabla_u \left[ (\phi(x + u))^\top Jv \right]_{u=0} = \left[ \left( \frac{d\phi}{du}(x + u) \right)^\top Jv + (\phi(x + u))^\top \frac{d}{du} Jv \right]_{u=0}$$

$$= \left[ \left( \frac{d\phi}{du}(x + u) \right)^\top Jv + (\phi(x + u))^\top \mathbf{0} \right]_{u=0}$$

$$= \left( \frac{d\phi}{du}(x) \right)^\top Jv = J^\top Jv$$

This allows us to efficiently approximate the solution of (12) to obtain $(J^\top J)^{-1} \nabla \hat{f}$. We use 5 iterations of the conjugate gradient algorithm in practice.

From there, it easy to solve for $\lambda$, given that $(J^+)^\top \nabla \hat{f} = J(J^\top J)^{-1} \nabla \hat{f}$. Then, $\delta^*$ can be calculated via (10). See Algorithm 1 for the full attack.

**Computational complexity**    PPGD's running time scales with the number of steps $T$ and the number of conjugate gradient iterations $K$. It also depends on whether the attack is self-bounded (the same network is used for classification and the LPIPS distance) or externally-bounded (different networks are used).

For each of the $T$ steps, $\theta(\widetilde{\mathbf{x}})$, $\nabla_{\widetilde{\mathbf{x}}} \mathcal{L}(f(\widetilde{\mathbf{x}}), y)$, and $\phi(\widetilde{\mathbf{x}} + h\delta_k)$ must be calculated once (lines 4 and 15 in Algorithm 1). This takes 2 forward passes and 1 backward pass for the self-bounded case, and 3 forward passes and 1 backward pass for the externally-bounded case.

In addition, $J^\top Jv$ needs to be calculated (in the MULTIPLYJACOBIAN routine) $K + 1$ times. Each calculation of $J^\top Jv$ requires 1 forward and 1 backward pass, assuming $\phi(\widetilde{\mathbf{x}})$ is already calculated.

Finally, the projection step takes $n + 1$ forward passes for $n$ iterations of the bisection method (see

Section A.4).

In all, the algorithm requires $T(K + n + 4)$ forward passes and $T(K + n + 3)$ backward passes in the self-bounded case. In the externally-bounded case, it requires $T(K + n + 5)$ forward passes and the same number of backward passes.

---

**Algorithm 1** Perceptual PGD (PPGD)

1: **procedure** PPGD(classifier $f(\cdot)$, LPIPS network $\phi(\cdot)$, input $\mathbf{x}$, label $y$, bound $\epsilon$, step $\eta$)
2: $\quad \widetilde{\mathbf{x}} \leftarrow \mathbf{x} + 0.01 * \mathcal{N}(0, 1)$ $\qquad\qquad\qquad$ ▷ initialize perturbations with random Gaussian noise
3: $\quad$ **for** $t$ in $1, \ldots, T$ **do** $\qquad\qquad\qquad\qquad\qquad\qquad$ ▷ $T$ is the number of steps
4: $\qquad \nabla \hat{f} \leftarrow \nabla_{\widetilde{\mathbf{x}}} \mathcal{L}(f(\widetilde{\mathbf{x}}), y)$
5: $\qquad \delta_0 \leftarrow 0$
6: $\qquad r_0 \leftarrow \nabla \hat{f} - \text{MULTIPLYJACOBIAN}(\phi, \widetilde{\mathbf{x}}, \delta_0)$
7: $\qquad p_0 \leftarrow r_0$
8: $\qquad$ **for** $k$ in $0, \ldots, K - 1$ **do** $\qquad$ ▷ conjugate gradient algorithm; we use $K = 5$ iterations
9: $\qquad\qquad \alpha_k \leftarrow \frac{r_k^\top r_k}{p_k^\top \text{MULTIPLYJACOBIAN}(\phi, \widetilde{\mathbf{x}}, p_k)}$
10: $\qquad\qquad \delta_{k+1} \leftarrow \delta_k + \alpha_k p_k$
11: $\qquad\qquad r_{k+1} \leftarrow r_k - \alpha_k \text{MULTIPLYJACOBIAN}(\phi, \widetilde{\mathbf{x}}, p_k)$
12: $\qquad\qquad \beta_k \leftarrow \frac{r_{k+1}^\top r_{k+1}}{r_k^\top r_k}$
13: $\qquad\qquad p_{k+1} \leftarrow r_{k+1} + \beta_k p_k$
14: $\qquad$ **end for**
15: $\qquad m \leftarrow \|\phi(\widetilde{\mathbf{x}} + h\delta_k) - \phi(\widetilde{\mathbf{x}})\|/h$ $\qquad\qquad$ ▷ $m \approx \|J\delta_k\|$ for small $h$; we use $h = 10^{-3}$
16: $\qquad \widetilde{\mathbf{x}} \leftarrow (\eta/m)\delta_k$
17: $\qquad \widetilde{\mathbf{x}} \leftarrow \text{PROJECT}(d, \widetilde{\mathbf{x}}, \mathbf{x}, \epsilon)$
18: $\quad$ **end for**
19: $\quad$ **return** $\widetilde{\mathbf{x}}$
20: **end procedure**
21:
22: **procedure** MULTIPLYJACOBIAN($\phi(\cdot)$, $\widetilde{\mathbf{x}}$, $\mathbf{v}$) $\quad$ ▷ calculates $J^\top J\mathbf{v}$; $J$ is the Jacobian of $\phi$ at $\widetilde{\mathbf{x}}$
23: $\quad J\mathbf{v} \leftarrow (\phi(\widetilde{\mathbf{x}} + h\mathbf{v}) - \phi(\widetilde{\mathbf{x}}))/h$ $\qquad\qquad$ ▷ $h$ is a small positive value; we use $h = 10^{-3}$
24: $\quad J^\top J\mathbf{v} \leftarrow \nabla_{\mathbf{u}} \left[\phi(\widetilde{\mathbf{x}} + \mathbf{u})^\top J\mathbf{v}\right]_{\mathbf{u}=0}$
25: $\quad$ **return** $J^\top J\mathbf{v}$
26: **end procedure**

---

## A.2 LAGRANGIAN PERCEPTUAL ATTACK (LPA)

Our second attack, Lagrangian Perceptual Attack (LPA), optimizes a Lagrangian relaxation of the perceptual attack problem (3):

$$\max_{\widetilde{\mathbf{x}}} \quad \mathcal{L}(f(\widetilde{\mathbf{x}}), y) - \lambda \max\left(0, \|\phi(\widetilde{\mathbf{x}}) - \phi(\mathbf{x})\|_2 - \epsilon\right). \tag{13}$$

To optimize (13), we use a variation of gradient descent over $\widetilde{\mathbf{x}}$, starting at $\mathbf{x}$ with a small amount of noise added. We perform our modified version of gradient descent for $T$ steps. We use a step size $\eta$, which begins at $\epsilon$ and decays exponentially to $\epsilon/10$.

At each step, we begin by taking the gradient of (13) with respect to $\widetilde{\mathbf{x}}$; let $\Delta$ refer to this gradient. Then, we normalize $\Delta$ to have $L_2$ norm 1, i.e. $\hat{\Delta} = \Delta/\|\Delta\|_2$. We wish to take a step in the direction of $\hat{\Delta}$ of size $\eta$ in LPIPS distance. If we wanted to take a step of size $\eta$ in $L_2$ distance, we could just take the step $\eta\hat{\Delta}$. However, taking a step of particular size in LPIPS distance is harder. We assume that the LPIPS distance is approximately linear in the direction $\hat{\Delta}$. We can approximate the directional derivative of the LPIPS distance in the direction $\hat{\Delta}$ using finite differences:

$$\frac{d}{d\alpha} d(\widetilde{\mathbf{x}}, \widetilde{\mathbf{x}} + \alpha\hat{\Delta}) \approx \frac{d(\widetilde{\mathbf{x}}, \widetilde{\mathbf{x}} + h\Delta)}{h} = m.$$

Here, $h$ is a small positive value, and we assign the approximation of the directional derivative to $m$. Now, we can write the first-order Taylor expansion of the perceptual distance towards the direction

$\hat{\Delta}$ as follows:
$$d(\widetilde{\mathbf{x}}, \widetilde{\mathbf{x}} + \alpha\hat{\Delta}) \approx d(\widetilde{\mathbf{x}}, \widetilde{\mathbf{x}}) + m\alpha = m\alpha.$$
we want to take a step of size $\eta$. Plugging in and solving, we obtain
$$\eta = d(\widetilde{\mathbf{x}}, \widetilde{\mathbf{x}} + \alpha\hat{\Delta}) \approx m\alpha$$
$$\eta \approx m\alpha$$
$$\eta/m \approx \alpha.$$

So, the approximate step we should take is $(\eta/m)\hat{\Delta}$. We take this step at each of the $T$ iterations of our modified gradient descent method.

We begin with $\lambda = 10^{-2}$. After performing gradient descent, if $d(\mathbf{x}, \widetilde{\mathbf{x}}) > \epsilon$ (i.e. the adversarial example is outside the constraint) we increase $\lambda$ by a factor of 10 and repeat the optimization. We repeat this entire process five times, meaning we search over $\lambda \in \{10^{-2}, 10^{-1}, 10^0, 10^1, 10^2\}$. Finally, if the resulting adversarial example is still outside the constraint, we project it into the threat model; see Appendix 5.

**Computational complexity** LPA's running time scales with the number of iterations $S$ used to search for $\lambda$ as well as the number of gradient descent steps $T$. $\phi(\mathbf{x})$ may be calculated once during the entire attack, which speeds it up. Then, each step of gradient descent requires 2 forward and 1 backward passes in the self-bounded case, and 3 forward and 2 backward passes in the externally-bounded case.

The projection at the end of the attack requires $n + 1$ forward passes for $n$ iterations of the bisection method (see Section A.4).

In total, the attack requires $2ST + n + 2$ forward passes and $ST + n + 2$ backward passes in the self-bounded case, and $3ST + n + 2$ forward passes and $2ST + n + 2$ backward passes in the externally-bounded case.

---

**Algorithm 2** Lagrangian Perceptual Attack (LPA)

---

1: **procedure** LPA(classifier network $f(\cdot)$, LPIPS distance $d(\cdot, \cdot)$, input $\mathbf{x}$, label $y$, bound $\epsilon$)
2:     $\lambda \leftarrow 0.01$
3:     $\widetilde{\mathbf{x}} \leftarrow \mathbf{x} + 0.01 * \mathcal{N}(0, 1)$         $\triangleright$ initialize perturbations with random Gaussian noise
4:     **for** $i$ in $1, \ldots, S$ **do**         $\triangleright$ we use $S = 5$ iterations to search for the best value of $\lambda$
5:         **for** $t$ in $1, \ldots, T$ **do**         $\triangleright$ $T$ is the number of steps
6:             $\Delta \leftarrow \nabla_{\widetilde{\mathbf{x}}}\left[\mathcal{L}(f(\widetilde{\mathbf{x}}), y) - \lambda \max\left(0, d(\widetilde{\mathbf{x}}, \mathbf{x}) - \epsilon\right)\right]$   $\triangleright$ take the gradient of (5)
7:             $\hat{\Delta} = \Delta/\|\Delta\|_2$         $\triangleright$ normalize the gradient
8:             $\eta = \epsilon * (0.1)^{t/T}$         $\triangleright$ the step size $\eta$ decays exponentially
9:             $m \leftarrow d(\widetilde{\mathbf{x}}, \widetilde{\mathbf{x}} + h\hat{\Delta})/h$   $\triangleright$ $m \approx$ derivative of $d(\widetilde{\mathbf{x}}, \cdot)$ in the direction of $\hat{\Delta}$; $h = 0.1$
10:            $\widetilde{\mathbf{x}} \leftarrow \widetilde{\mathbf{x}} + (\eta/m)\hat{\Delta}$         $\triangleright$ take a step of size $\eta$ in LPIPS distance
11:         **end for**
12:         **if** $d(\widetilde{\mathbf{x}}, \mathbf{x}) > \epsilon$ **then**
13:             $\lambda \leftarrow 10\lambda$         $\triangleright$ increase $\lambda$ if the attack goes outside the bound
14:         **end if**
15:     **end for**
16:     $\widetilde{\mathbf{x}} \leftarrow \text{PROJECT}(d, \widetilde{\mathbf{x}}, \mathbf{x}, \epsilon)$
17:     **return** $\widetilde{\mathbf{x}}$
18: **end procedure**

---

### A.3 FAST LAGRANGIAN PERCEPTUAL ATTACK

We use the Fast Lagrangian Perceptual Attack (Fast-LPA) for Perceptual Adversarial Training (PAT, see Section 5). Fast-LPA is similar to LPA (Appendix A.2), with two major differences. First, Fast-LPA does not search over $\lambda$ values; instead, during the $T$ gradient descent steps, $\lambda$ is increased exponentially from 1 to 10. Second, we remove the projection step at the end of the attack. This means that Fast-LPA may produce adversarial examples outside the threat model. This means that Fast-LPA cannot be used for evaluation, but it is fine for training.

**Computational complexity** Fast-LPA's running time can be calculated similarly to LPA's (see Section A.2), except that $S = 1$ and there is no projection step. Let $T$ be the number of steps taken during the attack. Then Fast-LPA requires $2T + 1$ forward passes and $T + 1$ backward passes for the self-bounded case, and $3T + 1$ forward passes and $2T + 1$ backward passes for the externally-bounded case.

In comparison, PGD with $T$ iterations requires $T$ forward passes and $T$ backward passes. Thus, Fast-LPA is slightly slower, requiring $T + 1$ more forward passes and no more backward passes.

---

**Algorithm 3** Fast Lagrangian Perceptual Attack (Fast-LPA)

---

1: **procedure** FASTLPA(classifier network $f(\cdot)$, LPIPS distance $d(\cdot, \cdot)$, input $\mathbf{x}$, label $y$, bound $\epsilon$)
2:     $\widetilde{\mathbf{x}} \leftarrow \mathbf{x} + 0.01 * \mathcal{N}(0, 1)$                                   $\triangleright$ initialize perturbations with random Gaussian noise
3:     **for** $t$ in $1, \ldots, T$ **do**                                   $\triangleright$ $T$ is the number of steps
4:         $\lambda \leftarrow 10^{t/T}$                                     $\triangleright$ $\lambda$ increases exponentially
5:         $\Delta \leftarrow \nabla_{\widetilde{\mathbf{x}}} \left[ \mathcal{L}(f(\widetilde{\mathbf{x}}), y) - \lambda \max \left( 0, d(\widetilde{\mathbf{x}}, \mathbf{x}) - \epsilon \right) \right]$       $\triangleright$ take the gradient of (5)
6:         $\hat{\Delta} = \Delta / \|\Delta\|_2$                                   $\triangleright$ normalize the gradient
7:         $\eta = \epsilon * (0.1)^{t/T}$                          $\triangleright$ the step size $\eta$ decays exponentially
8:         $m \leftarrow d(\widetilde{\mathbf{x}}, \widetilde{\mathbf{x}} + h\hat{\Delta})/h$       $\triangleright$ $m \approx$ derivative of $d(\widetilde{\mathbf{x}}, \cdot)$ in the direction of $\hat{\Delta}$; $h = 0.1$
9:         $\widetilde{\mathbf{x}} \leftarrow \widetilde{\mathbf{x}} + (\eta/m)\hat{\Delta}$                      $\triangleright$ take a step of size $\eta$ in LPIPS distance
10:     **end for**
11:     **return** $\widetilde{\mathbf{x}}$
12: **end procedure**

---

### A.4 PERCEPTUAL PROJECTION

We explored two methods of projecting adversarial examples into the LPIPS thread model. The method we use throughout the paper is based on Newton's method and is shown in algorithm 4. However, we also experimented with the bisection root finding method, shown in Algorithm 5 (also see Appendix E).

In general, given an adversarial example $\widetilde{\mathbf{x}}$, original input $\mathbf{x}$, and LPIPS bound $\epsilon$, we wish to find a projection $\widetilde{\mathbf{x}}'$ of $\widetilde{\mathbf{x}}$ such that $d(\widetilde{\mathbf{x}}', \mathbf{x}) \leq \epsilon$. Assume for this section that $d(\widetilde{\mathbf{x}}, \mathbf{x}) > \epsilon$, i.e. the current adversarial example $\widetilde{\mathbf{x}}$ is outside the bound. If $d(\widetilde{\mathbf{x}}, \mathbf{x}) \leq \epsilon$, then we can just let $\widetilde{\mathbf{x}}' = \widetilde{\mathbf{x}}$ and be done.

**Newton's method** The second projection method we explored uses the generalized Newton–Raphson method to attempt to find the closest projection $\widetilde{\mathbf{x}}'$ to the current adversarial example $\widetilde{\mathbf{x}}$ such that the projection is within the threat model, i.e. $d(\widetilde{\mathbf{x}}', \mathbf{x}) \leq \epsilon$. To find such a projection, we again define a function $r(\cdot)$ and look for its roots:

$$r(\widetilde{\mathbf{x}}') = d(\widetilde{\mathbf{x}}', \mathbf{x}) - \epsilon.$$

If we can find a projection $\widetilde{\mathbf{x}}'$ close to $\widetilde{\mathbf{x}}$ such that $r(\widetilde{\mathbf{x}}') \leq 0$, then this projection will be contained within the threat model, since

$$r(\widetilde{\mathbf{x}}') \leq 0 \Rightarrow d(\widetilde{\mathbf{x}}', \mathbf{x}) \leq \epsilon.$$

To find such a root, we use the generalized Newton-Raphson method, an iterative algorithm. Beginning with $\widetilde{\mathbf{x}}'_0 = \widetilde{\mathbf{x}}$, we update $\widetilde{\mathbf{x}}'$ iteratively using the step

$$\widetilde{\mathbf{x}}'_{i+1} = \widetilde{\mathbf{x}}'_i - \left[ \nabla r(\widetilde{\mathbf{x}}'_i) \right]^+ \left( r(\widetilde{\mathbf{x}}'_i) + s \right),$$

where $A^+$ denotes the pseudoinverse of $A$, and $s$ is a small positive constant (the "overshoot"), which helps the algorithm converge. We continue this process until $r(\widetilde{\mathbf{x}}'_t) \leq 0$, at which point the projection is complete.

This algorithm usually takes 2-3 steps to converge with $s = 10^{-2}$. Each step requires 1 forward and 1 backward pass to calculate $r(\widetilde{\mathbf{x}}'_t)$ and its gradient. The method also requires 1 forward pass at the beginning to calculate $\phi(\mathbf{x})$.

---

**Algorithm 4** Perceptual Projection (Newton's Method)

**procedure** PROJECT(LPIPS distance $d(\cdot, \cdot)$, adversarial example $\widetilde{\mathbf{x}}$, original input $\mathbf{x}$, bound $\epsilon$)
    $\widetilde{\mathbf{x}}'_0$
    **for** $i$ in $0, \ldots$ **do**
        $r(\widetilde{\mathbf{x}}'_i) \leftarrow d(\widetilde{\mathbf{x}}'_i, \mathbf{x}) - \epsilon$
        **if** $r(\widetilde{\mathbf{x}}'_i) \leq 0$ **then**
            **return** $\widetilde{\mathbf{x}}'_i$
        **end if**
        $\widetilde{\mathbf{x}}'_{i+1} = \widetilde{\mathbf{x}}'_i - \left[\nabla r(\widetilde{\mathbf{x}}'_i)\right]^+ \left(r(\widetilde{\mathbf{x}}'_i) + s\right)$          $\triangleright$ $s$ is the "overshoot"; we use $s = 10^{-2}$
    **end for**
**end procedure**

---

**Bisection method**      The first projection method we explored (and the one we use throughout the paper) attempts to find a projection $\widetilde{\mathbf{x}}'$ along the line connecting the current adversarial example $\widetilde{\mathbf{x}}$ and original input $\mathbf{x}$. Let $\delta = \widetilde{\mathbf{x}} - \mathbf{x}$. Then we can represent our final projection $\widetilde{\mathbf{x}}'$ as a point between $\mathbf{x}$ and $\widetilde{\mathbf{x}}$ as

$$\widetilde{\mathbf{x}}' = \mathbf{x} + \alpha\delta,$$

for some $\alpha \in [0, 1]$. If $\alpha = 0$, $\widetilde{\mathbf{x}}' = \mathbf{x}$; if $\alpha = 1$, $\widetilde{\mathbf{x}}' = \widetilde{\mathbf{x}}$. Now, define a function $r : [0, 1] \to \mathbb{R}$ as

$$r(\delta) = d(\mathbf{x} + \alpha\delta, \mathbf{x}) - \epsilon.$$

This function has the following properties:

1. $r(0) < 0$, since $r(0) = d(\mathbf{x}, \mathbf{x}) - \epsilon = -\epsilon$.
2. $r(1) > 0$, since $r(1) = d(\widetilde{\mathbf{x}}, \mathbf{x}) - \epsilon > 0$ because $d(\widetilde{\mathbf{x}}, \mathbf{x}) > \epsilon$.
3. $r(\alpha) = 0$ iff $d(\widetilde{\mathbf{x}}', \mathbf{x}) = \epsilon$.

We use the bisection root finding method to find a root $\alpha^*$ of $r(\cdot)$ on the interval $[0, 1]$, which exists since $r(\cdot)$ is continuous and because of items 1 and 2 above. By item 3, at this root, the projected adversarial example is within the threat model:

$$d(\widetilde{\mathbf{x}}', \mathbf{x}) = d(\mathbf{x} + \alpha^*\delta, \mathbf{x}) = r(\alpha^*) + \epsilon = \epsilon$$

We use $n = 10$ iterations of the bisection method to calculate $\alpha^*$. This requires $n + 1$ forward passes through the LPIPS network, since $\phi(\mathbf{x})$ must be calculated once, and $\phi(\mathbf{x} + \alpha\delta)$ must be calculated $n$ times. See Algorithm 5 for the full projection algorithm.

---

**Algorithm 5** Perceptual Projection (Bisection Method)

**procedure** PROJECT(LPIPS distance $d(\cdot, \cdot)$, adversarial example $\widetilde{\mathbf{x}}$, original input $\mathbf{x}$, bound $\epsilon$)
    $\alpha_{\min}, \alpha_{\max} \leftarrow 0, 1$
    $\delta \leftarrow \widetilde{\mathbf{x}} - \mathbf{x}$
    **for** $i$ in $1, \ldots, n$ **do**
        $\alpha \leftarrow (\alpha_{\min} + \alpha_{\max})/2$
        $\widetilde{\mathbf{x}}' \leftarrow \mathbf{x} + \alpha\delta$
        **if** $d(\mathbf{x}, \widetilde{\mathbf{x}}') > \epsilon$ **then**
            $\alpha_{\max} \leftarrow \alpha$
        **else**
            $\alpha_{\min} \leftarrow \alpha$
        **end if**
    **end for**
    **return** $\widetilde{\mathbf{x}}'$
**end procedure**

---

# B   ADDITIONAL RELATED WORK

Here, we expand on the related work discussed in Section 2 discuss some additional existing work on adversarial robustness.

**Adversarial attacks** Much of the initial work on adversarial robustness focused on perturbations to natural images which were bounded by the $L_2$ or $L_\infty$ distance (Carlini and Wagner, 2017; Goodfellow et al., 2015; Madry et al., 2018). However, recently the community has discovered many other types of perturbations that are imperceptible and can be optimized to fool a classifier, but are outside $L_p$ threat models. These include spatial perturbations using flow fields (Xiao et al., 2018), translation and rotation (Engstrom et al., 2017), and Wassterstein distance bounds (Wong et al., 2019). Attacks that manipulate the colors in images uniformly also been proposed (Hosseini and Poovendran, 2018; Hosseini et al., 2017; Zhang et al., 2019b) and have been generalized into "functional adversarial attacks" by Laidlaw and Feizi (2019).

A couple papers have proposed adversarial threat models that do not focus on a simple, manually defined perturbation type. Dunn et al. (2020) use a generative model of images; they perturb the features at various layers in the generator to create adversarial examples. Xu et al. (2020) train an autoencoder and then perturb images in representation space rather than pixel space.

## C ADDITIONAL ADVERSARIAL EXAMPLES

| Original | Self-bd. PPGD | External-bd. PPGD | Self-bd. LPA | External-bd. LPA |
|---|---|---|---|---|

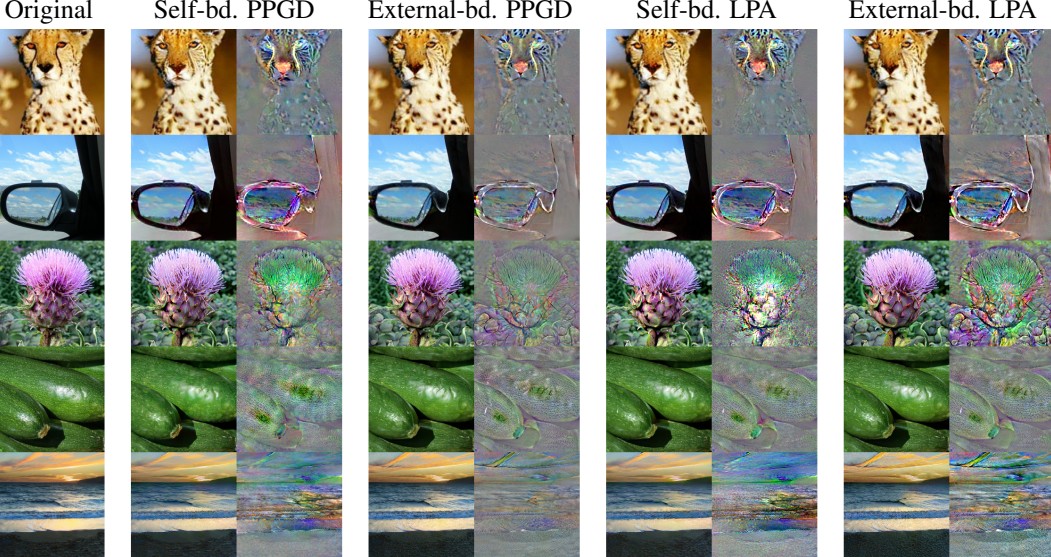

Figure 7: Adversarial examples generated using self-bounded and externally-bounded PPGD and LPA perceptual adversarial attacks (Section 4) with a large bound. Original images are shown in the left column and magnified differences from the original are shown to the right of the examples.

## D  ADDITIONAL PERCEPTUAL STUDY RESULTS

Table 4: Bounds and results from the perceptual study. Each threat model was evaluated with a small, medium, and large bound. Bounds for $L_2$, $L_\infty$, and JPEG attacks (first three rows) are given assuming input image is in the range $[0, 255]$. Perceptibility (perc.) is the proportion of natural input-adversarial example pairs annotated as "different" by participants. Strength (str.) is the success rate when attacking a classifier adversarially trained against that threat model (higher is stronger). Perceptual attacks (PPGD and LPA, see Section 4) are externally bounded with AlexNet. All experiments on ImageNet-100.

| Threat model | Small bound | | | Medium bound | | | Large bound | | |
|---|---|---|---|---|---|---|---|---|---|
| | Bound | Perc. | Str. | Bound | Perc. | Str. | Bound | Perc. | Str. |
| $L_\infty$ | 2 | 7% | 30.4% | 4 | 11% | 44.3% | 8 | 39% | 55.7% |
| $L_2$ | 600 | 8% | 40.6% | 1200 | 12% | 59.0% | 2400 | 51% | 88.5% |
| **JPEG-$L_\infty$** | 0.0625 | 6% | 19.5% | 0.125 | 16% | 25.2% | 0.25 | 51% | 44.0% |
| **StAdv** | 0.025 | 8% | 28.1% | 0.05 | 17% | 34.7% | 0.1 | 37% | 55.4% |
| **ReColorAdv** | 0.03 | 14% | 14.8% | 0.06 | 42% | 30.8% | 0.12 | 75% | 71.1% |
| **PPGD** | 0.25 | 7% | 47.9% | 0.5 | 9% | 68.6% | 1 | 25% | 91.8% |
| **LPA** | 0.25 | 14% | 76.5% | 0.5 | 27% | 98.4% | 1 | 45% | 100.0% |

## E  PERCEPTUAL ATTACK EXPERIMENTS

We experiment with variations of the two validation attacks, PPGD and LPA, described in Section 4. As described in Appendix A.4, we developed two methods for projecting candidate adversarial examples into the LPIPS ball surrounding a natural input. We attack a single model using PPGD and LPA with both projection methods. We also compare self-bounded to externally-bounded attacks.

We find that LPA tends to be more powerful than PPGD. Finally, we note that externally-bounded LPA is extremely powerful, reducing the accuracy of a PAT-trained classifier on ImageNet-100 to just 2.4%.

Besides these experiments, we always use externally-bounded attacks with AlexNet for evaluation. AlexNet correlates with human perception of adversarial examples (Figure 6) and provides a standard measure of LPIPS distance; in contrast, self-bounded attacks by definition have varying bounds across evaluated models.

Table 5: Accuracy of a PAT-trained ResNet-50 on ImageNet-100 against various perceptual adversarial attacks. PPGD and LPA attacks are shown self-bounded and externally-bounded with AlexNet. We also experimented with two different perceptual projection methods (see Appendix A.4). Bounds are $\epsilon = 0.25$ for self-bounded attacks and $\epsilon = 0.5$ for externally-bounded attacks, since the LPIPS distance from AlexNet tends to be about twice as high as that from ResNet-50.

| Attack | LPIPS model | Projection | Accuracy against PAT |
|---|---|---|---|
| PPGD | self | bisection method | 43.3 |
| PPGD | self | Newton's method | 49.7 |
| PPGD | AlexNet | bisection method | 45.2 |
| PPGD | AlexNet | Newton's method | 28.7 |
| LPA | self | bisection method | 39.6 |
| LPA | self | Newton's method | 53.8 |
| LPA | AlexNet | bisection method | 4.2 |
| LPA | AlexNet | Newton's method | 2.4 |

## F  PAT Experiments

### F.1  Ablation Study

We perform an ablation study of Perceptual Adversarial Training (PAT). First, we examine Fast-LPA, the training attack. We attempt training without step size ($\eta$) decay and/or without increasing $\lambda$ during Fast-LPA, and find that PAT performs best with both $\eta$ decay and $\lambda$ increase.

Training a classifier with PAT gives robustness against a wide range of adversarial threat models (see Section 7). However, it tends to give low accuracy against natural, unperturbed inputs. Thus, we use a technique from Balaji et al. (2019) to improve natural accuracy in PAT-trained models: at each training step, only inputs which are classified correctly without any perturbation are attacked. In addition to increasing natural accuracy, this also improves the speed of PAT since only some inputs from each batch must be attacked. In this ablation study, we compare attacking every input with Fast-LPA during training to only attacking the natural inputs which are already classified correctly. We find that the latter method achieves higher natural accuracy at the cost of some robust accuracy.

Table 6: Accuracies against various attacks for models in the PAT ablation study. Attack bounds are $8/255$ for $L_\infty$, 1 for $L_2$, 0.5 for PPGD/LPA, and the original bounds for StAdv/ReColorAdv.

| Ablation | Union | Unseen mean | Narrow threat models | | | | | NPTM | |
|---|---|---|---|---|---|---|---|---|---|
| | | | Clean | $L_\infty$ | $L_2$ | StAdv | ReColor | PPGD | LPA |
| None | 21.9 | 45.6 | 82.4 | 30.2 | 34.9 | 46.4 | 71.0 | 13.1 | 2.1 |
| No $\eta$ decay | 17.2 | 48.1 | 82.7 | 37.5 | 43.3 | 39.7 | 72.1 | 10.5 | 1.1 |
| No $\lambda$ increase | 8.1 | 42.1 | 85.1 | 33.7 | 35.9 | 27.8 | 71.1 | 11.6 | 1.0 |
| No $\eta$ decay or $\lambda$ increase | 19.6 | 49.2 | 82.1 | 36.7 | 41.9 | 44.8 | 73.4 | 10.9 | 0.9 |
| Attack all inputs | 26.8 | 46.6 | 74.5 | 29.8 | 33.5 | 56.6 | 66.4 | 24.5 | 6.5 |

### F.2  Projection During Training

We choose not to add a projection step to the end of Fast-LPA during training because it slows down the attack, requiring many more passes through the network per training step. However, we tested self-bounded PAT with a projection step and found that it increased clean accuracy slightly but decreased robust accuracy significantly. We believe this is because not projecting increases the effective bound on the training attacks, leading to better robustness. To test this, we tried training without projection using a smaller bound ($\epsilon = 0.4$ instead of $\epsilon = 0.5$) and found the results closely matched the results when using projection at the larger bound. That is, PAT with projection at $\epsilon = 0.5$ is similar to PAT without projection at $\epsilon = 0.4$. These results are shown in 7.

Table 7: Accuracies against various attacks for PAT-trained models on CIFAR-10, with and without a projection step during training.

| Projection | Bound ($\epsilon$) | Union | Unseen mean | Narrow threat models | | | | | NPTM | |
|---|---|---|---|---|---|---|---|---|---|---|
| | | | | Clean | $L_\infty$ | $L_2$ | StAdv | ReColor | PPGD | LPA |
| No | 0.5 | 21.9 | 45.6 | 82.4 | 30.2 | 34.9 | 46.4 | 71.0 | 13.1 | 2.1 |
| Yes | 0.5 | 5.8 | 41.4 | 83.9 | 35.7 | 38.5 | 17.9 | 73.3 | 10.7 | 1.7 |
| No | 0.4 | 4.9 | 36.3 | 84.1 | 31.1 | 36.3 | 10.3 | 67.4 | 10.9 | 3.2 |

### F.3  Self-Bounded vs. AlexNet-Bounded PAT

Performing externally-bounded PAT with AlexNet produces more robust models on CIFAR-10 than self-bounded PAT. This is not the case on ImageNet-100, where self- and AlexNet-bounded PAT perform similarly.

There is a simple explanation for this: the effective training bound on CIFAR-10 is greater for AlexNet-bounded PAT than for self-bounded PAT. To measure this, we generate adversarial examples

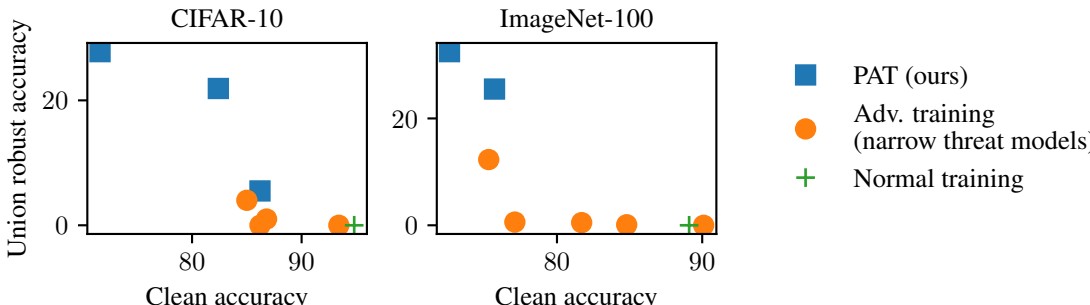

Figure 8: Several classifiers trained with PAT, adversarial training, and normal training on CIFAR-10 and ImageNet-100 are plotted with their clean accuracy and accuracy against the union of narrow threat models (see Section 7 for robustness evaluation methodology). PAT models on both datasets outperform adversarial trained models in both clean and robust accuracy.

for all the test threat models on CIFAR-10 ($L_2$, $L_\infty$, StAdv, and ReColorAdv). We find that the average LPIPS distance for all adversarial examples using AlexNet is 1.13; for a PAT-trained ResNet-50, it is 0.88. Because of this disparity, we use a lower training bound for self-bounded PAT ($\epsilon = 0.5$) than for AlexNet-bounded PAT ($\epsilon = 1$). However, this means that the average test attack has 76% greater LPIPS distance than the training attacks for self-bounded PAT, whereas the average test attack has only 13% greater LPIPS distance for AlexNet-bounded PAT. This explains why AlexNet-bounded PAT gives better robustness; it only has to generalize to slightly larger attacks on average.

We tried performing AlexNet-bounded PAT with a more comparable bound ($\epsilon = 0.7$) to self-bounded PAT. This gives the average test attack about 80% greater LPIPS distance than the training attacks, similar to self-bounded PAT. Table 8 shows that the results are more similar for self-bounded and AlexNet-bounded PAT with $\epsilon = 0.7$.

Table 8: Accuracies against various attacks for classifiers on CIFAR-10 trained with self- and AlexNet-bounded PAT using various bounds.

| LPIPS model | Bound ($\epsilon$) | Union | Unseen mean | Clean | $L_\infty$ | $L_2$ | StAdv | ReColor | PPGD | LPA |
|---|---|---|---|---|---|---|---|---|---|---|
| | | | | | | | | | **NPTM** | |
| Self | 0.5 | 21.9 | 45.6 | 82.4 | 30.2 | 34.9 | 46.4 | 71.0 | 13.1 | 2.1 |
| AlexNet | 0.7 | 16.7 | 45.3 | 80.0 | 35.5 | 41.3 | 33.2 | 71.5 | 17.8 | 4.9 |
| AlexNet | 1.0 | 27.8 | 48.5 | 71.6 | 28.7 | 33.3 | 64.5 | 67.5 | 26.6 | 9.8 |

### F.4 ACCURACY-ROBUSTNESS TRADEOFF

Tsipras et al. (2019) have noted that there is often a tradeoff the between adversarial robustness of a classifier and its accuracy. That is, models which have higher accuracy under adversarial attack may have lower accuracy against clean images. We observe this phenomenon with adversarial training and PAT. Since PAT gives greater robustness against several narrow threat models, models trained with it tend to have lower accuracy on clean images than models trained with narrow adversarial training. In Figure 8, we show the robust and clean accuracies of several models trained on CIFAR-10 and ImageNet-100 with PAT and adversarial training. While some PAT models have lower clean accuracy than adversarially trained models, at least one PAT model on each dataset surpasses the Pareto frontier of the accuracy-robustness tradeoff for adversarial training. That is, there are PAT-trained models on both datasets with both higher robust accuracy *and* higher clean accuracy than adversarial training.

### F.5 PERFORMANCE AGAINST STADV AND RECOLORADV

It was surprising to find that PAT outperformed threat-specific adversarial training (AT) against the StAdv and ReColorAdv attacks on CIFAR-10 (it does not do so on ImageNet-100). In Table 2 (partially reproduced in Table 9 below), PAT-AlexNet improves robustness over AT against StAdv from 54% to 65%; PAT-self improves robustness over AT against ReColorAdv from 65% to 71%.

We conjecture that, for these threat models, this is because training against a wider set of perturbations at training time helps generalize robustness to new inputs at test time, even within the same threat model. To test this, we additionally train classifiers using adversarial training against the StAdv and ReColorAdv attacks with *double* the default bound. The results are shown in Table 9 below. We find that, because these classifiers are exposed to a wider range of spatial and recoloring perturbations during training, they perform better than PAT against those attacks at test time (76% vs 65% for StAdv and 81% vs 71% for ReColorAdv).

This suggests that PAT not only improves robustness against a wide range of adversarial threat models, it can actually improve robustness over threat-specific adversarial training by incorporating a wider range of attacks during training.

Table 9: Results of our experiments training against StAdv and ReColorAdv on CIFAR-10 with double the default bounds. Columns are identical to Table 2.

| Training | Union | Unseen mean | Narrow threat models | | | | | NPTM | |
|---|---|---|---|---|---|---|---|---|---|
| | | | Clean | $L_\infty$ | $L_2$ | StAdv | ReColor | PPGD | LPA |
| AT StAdv | 0.0 | 1.4 | 86.2 | 0.1 | 0.2 | 53.9 | 5.1 | 0.0 | 0.0 |
| AT StAdv (double $\epsilon$) | 0.2 | 3.7 | 83.9 | 0.2 | 0.5 | 76.1 | 13.9 | 0.0 | 0.0 |
| AT ReColorAdv | 0.0 | 3.1 | 93.4 | 8.5 | 3.9 | 0.0 | 65.0 | 0.1 | 0.0 |
| AT ReColorAdv (double $\epsilon$) | 0.0 | 5.3 | 92.0 | 12.5 | 8.6 | 0.3 | 81.2 | 0.4 | 0.0 |
| PAT-self | 21.9 | 45.6 | 82.4 | 30.2 | 34.9 | 46.4 | 71.0 | 13.1 | 2.1 |
| PAT-AlexNet | 27.8 | 48.5 | 71.6 | 28.7 | 33.3 | 64.5 | 67.5 | 26.6 | 9.8 |

## G  COMMON CORRUPTIONS EVALUATION

We evaluate the robustness of PAT-trained models to common corruptions in addition to adversarial examples on CIFAR-10 and ImageNet-100. In particular, we test PAT-trained classifiers on CIFAR-10-C and ImageNet-100-C, where ImageNet-100-C is the 100-class subset of ImageNet-C formed by taking every tenth class (Hendrycks and Dietterich, 2019). These datasets are based on random corruptions of CIFAR-10 and ImageNet, respectively, using 15 perturbation types with 5 levels of severity. The perturbation types are split into four general categories: "noise," "blur," "weather," and "digital."

The metric we use to evaluate PAT against common corruptions is mean relative corruption error (relative mCE). The relative corruption error is defined by Hendrycks and Dietterich (2019) for a classifier $f$ and corruption type $c$ as

$$\text{Relative CE}_c^f = \frac{\sum_{s=1}^{5} E_{s,c}^f - E_{\text{clean}}^f}{\sum_{s=1}^{5} E_{s,c}^{\text{AlexNet}} - E_{\text{clean}}^{\text{AlexNet}}}$$

where $E_{s,c}^f$ is the error of classifier $f$ against corruption type $c$ at severity level $s$, and $E_{\text{clean}}^f$ is the error of classifier $f$ on unperturbed inputs. The relative mCE is defined as the mean relative CE over all perturbation types.

The relative mCE for classifiers trained with normal training, adversarial training, and PAT is shown in Tables 10 and 11. PAT gives better robustness (lower relative mCE) against common corruptions on both CIFAR-10-C and ImageNet-100-C. The only category of perturbations where $L_2$ adversarial training outperforms PAT is "noise" on CIFAR-10-C, which makes sense because Gaussian and other types of noise are symmetrically distributed in an $L_2$ ball. For the other perturbation types and on ImageNet-100-C, PAT outperforms $L_2$ and $L_\infty$ adversarial training, indicating that robustness against a wider range of worst-case perturbations also gives robustness against a wider range of random perturbations.

Table 10: Robustness of classifiers trained with adversarial training and PAT against common corruptions in the CIFAR-10-C dataset. Results are reported as relative mCE (lower is better). PAT improves robustness against common corruptions overall and for three of the specific perturbation categories.

| | Perturbation Type | | | | All |
|---|---|---|---|---|---|
| **Training** | Noise | Blur | Weather | Digital | |
| Normal | 1.72 | 1.57 | 1.03 | 1.87 | 1.59 |
| $L_\infty$ | 0.24 | 0.46 | 0.96 | 0.61 | 0.57 |
| $L_2$ | **0.16** | 0.39 | 1.02 | 0.61 | 0.54 |
| PAT-self | 0.22 | **0.30** | 0.90 | 0.58 | 0.50 |
| PAT-AlexNet | 0.22 | 0.34 | **0.88** | **0.56** | **0.49** |

Table 11: Robustness of classifiers trained with adversarial training and PAT against common corruptions in the ImageNet-100-C dataset. Results are reported as relative mCE (lower is better).

| | Perturbation Type | | | | All |
|---|---|---|---|---|---|
| **Training** | Noise | Blur | Weather | Digital | |
| Normal | 0.85 | 0.57 | **0.56** | 0.35 | 0.55 |
| $L_\infty$ | 0.54 | 0.41 | 0.62 | 0.25 | 0.42 |
| $L_2$ | 0.41 | 0.36 | 0.71 | 0.27 | 0.41 |
| PAT-self | **0.39** | 0.36 | 0.60 | **0.24** | **0.37** |
| PAT-AlexNet | 0.49 | **0.35** | 0.61 | **0.24** | 0.39 |

## H  EXPERIMENT DETAILS

For all experiments, we train ResNet-50 (He et al., 2016) with SGD for 100 epochs. We use 10 attack iterations for training and 200 for testing, except for PPGD and LPA, where we use 40 for testing since they are more expensive. Self-bounded PAT takes about 12 hours to train for CIFAR-10 on an Nvidia RTX 2080 Ti GPU, and about 5 days to train for ImageNet-100 on 4 GPUs. We implement PPGD, LPA, and PAT using PyTorch (Paszke et al., 2017).

We preprocess images after adversarial perturbation, but before classification, by standardizing them based on the mean and standard deviation of each channel for all images in the dataset. We use the default data augmentation techniques from the `robustness` library (Engstrom et al., 2019). The CIFAR-10 dataset can be obtained from `https://www.cs.toronto.edu/~kriz/cifar.html`. The ImageNet-100 dataset is a subset of the ImageNet Large Scale Visual Recognition Challenge (2012) (Russakovsky et al., 2015) including only every tenth class by WordNet ID order. It can be obtained from `http://www.image-net.org/download-images`.

Table 12: Hyperparameters for the adversarial training experiments on CIFAR-10 and ImageNet-100. For CIFAR-10, hyperparameters are similar to those used by Zhang et al. (2019a). For ImageNet-100, hyperparameters are similar to those used by Kang et al. (2019).

| Parameter | CIFAR-10 | ImageNet-100 |
|---|---|---|
| Architecture | ResNet-50 | ResNet-50 |
| Number of parameters | 23,520,842 | 23,712,932 |
| Optimizer | SGD | SGD |
| Momentum | 0.9 | 0.9 |
| Weight decay | $2 \times 10^{-4}$ | $10^{-4}$ |
| Batch size | 50 | 128 |
| Training epochs | 100 | 90 |
| Initial learning rate | 0.1 | 0.1 |
| Learning rate drop epochs ($\times 0.1$ drop) | 75, 90 | 30, 60, 80 |
| Attack iterations (train) | 10 | 10 |
| Attack iterations (test) | 200 | 200 |

## H.1 LAYERS FOR LPIPS CALCULATION

Calculating the LPIPS distance using a neural network classifier $g(\cdot)$ requires choosing layers whose normalized, flattened activations $\phi(\cdot)$ should be compared between images. For AlexNet and VGG-16, we use the same layers to calculate LPIPS distance as do Zhang et al. (2018). For AlexNet (Krizhevsky et al., 2012), we use the activations after each of the first five ReLU functions. For VGG-16 (Simonyan and Zisserman, 2014), we use the activations directly before the five max pooling layers. In ResNet-50, we use the outputs of the conv2_x, conv3_x, conv4_x, and conv5_x layers, as listed in Table 1 of He et al. (2016).

