# OpenReview forum: "Perceptual Adversarial Robustness: Defense Against Unseen Threat Models"
_ICLR.cc/2021/Conference — ICLR 2021 Poster_

### Official Review · AnonReviewer3 · 2020-10-16
**Interesting paper that solves an important problem**

**Rating:** 7
**Confidence:** 5

**Review:**

This paper proposes a threat model called Neural Perceptual Threat Model (NPTM) and under NPTM, they develop novel perceptual adversarial attacks: perceptual Projected Gradient Descent (PPGD) and Lagrangian Perceptual Attack (LPA). Also, they propose Perceptual Adversarial Training (PAT) which achieves good robustness against various types of adversarial attacks and even could generalize well to unforeseen perturbation types.

I think this is an interesting paper and solves an important problem. The writing is very clear and easy to follow. The main contributions are:

1. Introduce a new threat model called the Neural Perceptual Threat Model (NPTM) based on Learned Perceptual Image Patch Similarity (LPIPS) and propose two perceptual attack methods PPGD and LPA. They perform a study on Amazon Mechanical Turk (AMT) to show that LPIPS correlates well with human judgments across 7 different types of adversarial perturbations and adversarial examples generated by their attacks are imperceptible to humans. Such a study can benefit future research and they promise to release their dataset with annotations. They also perform experiments to show that LPA is by far the strongest adversarial attack at a given level of perceptibility, which could be used to evaluate the robustness of the defenses in the future.

2. Propose Perceptual Adversarial Training (PAT) which achieves state-of-the-art robustness against the union of the five attacks $L_2$, $L_\infty$, spatial, recoloring, and JPEG, without training against any of them. Being able to generalize well to unforeseen perturbation types is a desirable property for adversarial robustness and previous adversarial defenses don’t have such a property.

However, I have some concerns:

1. I notice that the clean accuracy of PAT-self and PAT-AlexNet (in Table 2&3) is lower than other training methods. I think it is due to removing the projection step at the end of the Fast-LPA attack. Could the authors explain why they need to remove the projection step? What are the results without removing the projection step? I think achieving good clean accuracy is also important for an adversarial trained model and PAT should not hurt the clean accuracy much;

2. Could the authors explain why AlexNet is the best proxy for human judgments of perceptual distance? Are there any insights? Also, could the authors provide detailed descriptions about how the AlexNet model is trained?

3. The authors mention that training against multiple threat models simultaneously will result in lower robustness against any one of the threat models when compared to hardening against that threat model alone. But from the experimental results, PAT is also less robust against a threat model compared to the method that trains against that threat model. Could the authors compare PAT to those methods that train against multiple threat models (e.g. [1] and [2])?

4. It would be good that the authors could evaluate PAT on images with common corruptions (e.g. CIFAR-10-C and ImageNet-C proposed in [3]). It can demonstrate that PAT could generalize well to unforeseen perturbation types further.

If the authors could address these concerns, I am willing to raise my scores.

Minor things: I find that the clean accuracy of the Normal model is 0 in Table 2. I think it is a typo. Please correct it.

[1] Maini, Pratyush, Eric Wong, and J. Zico Kolter. "Adversarial robustness against the union of multiple perturbation models." arXiv preprint arXiv:1909.04068 (2019).

[2] Tramèr, Florian, and Dan Boneh. "Adversarial training and robustness for multiple perturbations." Advances in Neural Information Processing Systems. 2019.

[3] Hendrycks, Dan, and Thomas Dietterich. "Benchmarking neural network robustness to common corruptions and perturbations." arXiv preprint arXiv:1903.12261 (2019).

---

> ### Author Response · Authors · 2020-11-18
> **Responses to your concerns 1 and 2**
>
> Thank you for your insightful and extensive review. Here are responses to your concerns:
>
>  1. We also believe achieving good clean accuracy is important for a robust model. In fact, we already use a technique in the paper to improve clean accuracy, described in Appendix F.1: we only attack inputs during training which are already classified correctly. This improves clean accuracy substantially on CIFAR-10. We also tested your suggestion to use the projection step during PAT, and found that this did not improve clean accuracy as much, but it significantly hurt robust accuracy. We believe this is because not projecting increases the effective bound on the training attacks, leading to better robustness. To test this, we tried training without projection using a smaller bound ($\epsilon = 0.4$ instead of $\epsilon = 0.5$) and found the results closely matched the results when using projection at the larger bound. See the table below for a comparison. We originally chose not to use a projection step during PAT because it significantly slows down training, requiring many more passes through the neural network for each training step. However, it turns out that not projecting further improves the robustness by increasing the effective attack bounds during the training. We have included this discussion along with evaluation of PAT with and without projection in Appendix F.2.
>
>    |Training method|Union$\quad$|Unseen Mean$\quad$|Clean$\quad$|$L_\infty$$\quad$|$L_2$$\quad$|StAdv$\quad$|ReColorAdv$\quad$|PPGD$\quad$|LPA$\quad$
>    |--|--|--|--|--|--|--|--|--|--|
>    |PAT-self (attack only correct)|21.9|45.6|82.4|30.2|34.9|46.4|71.0|13.1|2.1|
>    |PAT-self (attack every input)|26.8|46.6|74.5|29.8|33.5|56.6|66.4|24.5|6.5|
>    |PAT-self with projection|5.8|41.4|83.9|35.7|38.5|17.9|73.3|10.7|1.7|
>    |PAT-self with smaller bound ($\epsilon=0.4$)$\quad$|4.9|36.3|84.1|31.1|36.3|10.3|67.4|10.9|3.2|
>
>   2. We find that LPIPS distances based on both AlexNet and ResNet-50 provide good proxies for human perceptual distance; the difference between them is minor (see Figure 4c). In fact, even untrained AlexNet and ResNet-50 models
> perform as well or better than the $L_2$ distance; similar results are reported by the original LPIPS paper [1]. This is important for the beginning of self-bounded PAT because it ensures that reasonable adversarial examples are being generated even before the network has learned much.
>
>    We believe that part of the explanation for this surprising phenomenon is the inductive bias of convolution neural networks, which are known to have similarities to the human visual processing system [2]. Zhang et al. [1] also suggest that "perceptual similarity is an emergent property shared across deep visual representations." That is, CNNs and humans may induce comparable perceptual distances because they are both learning representations to solve similar visual tasks.
>
>    You also ask how we train AlexNet. For ImageNet-100, we simply use the pretrained PyTorch AlexNet model, as did the original LPIPS paper [1]. Even though this is trained on full ImageNet, we only use the internal representations and not the classifier outputs, so it works fine on ImageNet-100. On CIFAR-10, we train AlexNet using the same hyperparameters as for ResNet-50, except we use 1/10th the learning rate. We found that AlexNet trained on ImageNet did not work well for CIFAR-10 because of the vast difference in image resolution between the two datasets.
>
> Sorry about the typo in Table 2; we have corrected it.
>
> [1] Zhang et al. The Unreasonable Effectiveness of Deep Features as a Perceptual Metric, CVPR 2018.
>
> [2] Yamins and DiCarlo. Using goal-driven deep learning models to understand sensory cortex. Nature Neuroscience 19, 356–365 (2016)

---

> > ### Comment · AnonReviewer3 · 2020-11-18
> > **Thanks for the clarification**
> >
> > The authors' responses address some of my concerns. But I still concern about the low clean accuracy. I think there is a trade-off between clean accuracy and robust accuracy. A better discussion of it would benefit the paper. I keep the same rating due to this concern.

---

> > > ### Author Response · Authors · 2020-11-20
> > > **PAT improves on adversarial training in both clean and robust accuracy**
> > >
> > > We would appreciate it if you could elaborate on your comment on **low clean accuracy.** To be clear, our perceptual adversarial training (PAT) improves on **both** clean accuracy and adversarial robustness compared to L2 adversarial training on both CIFAR-10 and ImageNet-100 (see the tables below).
> > >
> > > Per your suggestion, we have added Appendix F.4 detailing the tradeoff between robustness and accuracy for PAT and adversarial training. We include a plot showing the tradeoff on both datasets in Figure 8.
> > >
> > > ### CIFAR-10
> > > |Training method|Clean|Union|Unseen Mean|
> > > |--|--|--|--|
> > > |AT $L_2$|85.0|4.0|25.3|
> > > |PAT-AlexNet ($\epsilon=0.5$)|**86.2**|**5.5**|**39.5**|
> > >
> > > ### ImageNet-100
> > > |Training method|Clean|Union|Unseen Mean|
> > > |--|--|--|--|
> > > |AT $L_2$|75.3|12.3|31.5|
> > > |PAT-AlexNet ($\epsilon=0.5$)|**75.7**|**25.5**|**44.7**|

---

> > > > ### Comment · AnonReviewer3 · 2020-11-20
> > > > **Concerns about clean accuracy**
> > > >
> > > > I mean the clean accuracy of PAT reported in Tables 2 and 3 is much lower than that of adversarially trained models, which is my concern.

---

> > > > ### Comment · AnonReviewer3 · 2020-11-22
> > > > **Thanks for the clarification!**
> > > >
> > > > The discussion in Appendix F.4 about the tradeoff between robustness and accuracy for PAT and adversarial training addresses my concern about low clean accuracy. Therefore, I raise my score to 7.

---

> ### Author Response · Authors · 2020-11-18
> **Responses to your concerns 3 and 4**
>
> 3. Between the submission and rebuttal, we compared PAT to the methods you mention which train against the union of multiple threat models. Specifically, we compare PAT to the "average" and "max" methods from Tramèr et al., as well as a "random" method which choose a threat model randomly at each training iteration. Results are given in the updated Tables 2 and 3, and the relevant part of Table 2 is reproduced below. PAT-self and PAT-AlexNet actually beat all three methods in union accuracy, *despite not training against any of the constituent threat models*. That is, PAT generalizes to unseen adversarial threat models.
>
>    |Training method$\quad$|Union$\quad$|Clean$\quad$|$L_\infty$$\quad$|$L_2$$\quad$|StAdv$\quad$|ReColorAdv$\quad$|PPGD$\quad$|LPA$\quad$|
>    |--|--|--|--|--|--|--|--|--|
>    |AT all (random)|0.7|85.2|22.0|23.4|1.2|46.9|1.8|0.1|
>    |AT all (average)|14.7|86.8|39.9|39.6|20.3|64.8|10.6|1.1|
>    |AT all (max)|21.4|84.0|25.7|30.5|40.0|63.8|8.6|1.1|
>    |PAT-self|21.9|82.4|30.2|34.9|46.4|71.0|13.1|2.1|
>    |PAT-AlexNet|**27.8**|71.6|28.7|33.3|64.5|67.5|**26.6**|**9.8**|
>
>   4. Thanks for the suggestion. We evaluated PAT against other methods on CIFAR-10-C. We also evaluated on ImageNet-100-C by taking the same subset of the classes (every 10th class) from ImageNet-C. Below are the results for normal training, adversarial training, and PAT, all using ResNet-50. PAT obtains improved (lower) relative mean corruption error (relative mCE) on both datasets compared to normal training and adversarial training. These results and more are presented in Appendix G in the revised paper.
>
>    |Training$\quad$|Relative mCE (CIFAR-10-C)$\quad$|Relative mCE (ImageNet-100-C)$\quad$|
>    |--|--|--|
>    |Normal|1.59|0.55|
>    |AT $L_\infty$|0.57|0.42|
>    |AT $L_2$|0.54|0.41|
>    |PAT-self|0.50|**0.37**|
>    |PAT-AlexNet$\quad$|**0.49**|0.39|

---

### Official Review · AnonReviewer1 · 2020-10-28

**Rating:** 6
**Confidence:** 4

**Review:**

This paper studies the adversarial robustness of deep neural networks against multiple and unforeseen threat models. Since there lacks a precise formalization of human perception, this paper adopts LPIPS, a metric that correlates well with human perception based on neural network activations. Then, two adversarial attack methods are proposed to generate adversarial examples under the metric. And an adversarial training method is also proposed. The experiments on various threat models validate the effectiveness of the proposed method.

The writing of this paper is clear. The generalizability of robust DNN against multiple threat models is an important problem. This paper is a good attempt to solve this problem. Based on a perception similarity metric, new adversarial attacks and defenses are studied. Thus the paper is comprehensive.

This biggest problem of this paper, in my opinion, is that the adopted metric is defined on neural networks. Although the authors have conducted human evaluations to prove that this metric correlates well with humans, I still doubt whether it can reflect all potential threats. For example, if the metric is perfect, any (adversarial) example, that is perceptually the same as a clean example to humans, could have the similar representation with the clean example when using the network \phi. If this is true, \phi itself is a perfectly robust network, and we do not need any adversarial training to robustify another network (i.e., f(x) in this paper).

However, defining a perfect metric for adversarial training may be very difficult. And this paper makes a step towards this goal.

---

> ### Author Response · Authors · 2020-11-18
> **The neural perceptual distance is not perfect, but it's better than $L_p$ norms**
>
> Thank you for your insightful review.
>
> You write that you "doubt whether [the neural perceptual distance] can reflect all potential threats." We agree that it is very difficult to define a perfect model of human perception, and we do not present our neural perceptual threat model (NPTM) as such. However, we believe our proposed threat model is a great improvement over using the $L_2$ or $L_\infty$ distances for adversarial training, which are much worse models of human perception. Furthermore, we have found that the LPIPS distance correlates well with human perception in all the diverse threat models we have explored, which include $L_p$-bounded, spatially-transformed, recolored, and JPEG-distorted adversarial examples. We pose as an open question and challenge to the adversarial robustness community to determine if there are imperceptible threat models which do not fall within the NPTM.
>
> You also remark that "if the metric is perfect, any (adversarial) example...could have the similar representation with the clean example when using the network $\phi$. If this is true, $\phi$ itself is a perfectly robust network." This is a sensible observation, and in fact, we argue that self-bounded PAT bootstraps itself towards producing such a network. At the beginning of self-bounded PAT, the network does not have any robustness. However, as PAT continues, the network $f$ is exposed to adversarial examples that robustify it; this in turn makes the neural perceptual distance $\phi$ based on it more robust. Thus, self-bounded PAT may be viewed as a way to learn both a robust neural perceptual distance and a robust classifier simultaneously. In this case, no robust perceptual metric is needed before training begins.
>
> Furthermore, having a robust perceptual distance does not necessarily translate into a robust classifier. Since the LPIPS distance depends on millions of internal activations of a network, attacking it is much more difficult than attacking the single classification output of the network; that is, classification is inherently less robust than the LPIPS distance. In fact, in self-bounded PAT, we explicitly try to find an adversarial example within a small perceptual distance that still fools the classifier, where the perceptual distance *is based on the same network used for classification*. That is, we find $\tilde{x}$ such that $\| \phi(x) - \phi(\tilde{x}) \| \leq \epsilon$, but $f(x) \neq f(\tilde{x})$. The fact that it is easy to find such an adversarial example $\tilde{x}$ shows that ensuring the representations for two similar images are always close does not ensure a robust classifier.

---

### Official Review · AnonReviewer4 · 2020-10-28
**ICLR 2021 Conference Paper261**

**Rating:** 7
**Confidence:** 4

**Review:**

Summary:
* This paper proposes perceptual adversarial robustness, an adversarial truing against the set of all imperceptible adversarial examples. Through a perceptual study, they approximate human perception with a neural neural - “neural perceptual distance”, and test the robustness against 5 threat models including L2 and Loo showing state-of-the-art robustness even without training on them, showing generality of the model considered.

Strengths:
* Relevant, challenging problem of interest to the CV and adversarial learning communities
* Clearly define scope and contributions
* Reasoning about both new robustness metric and new attacks to this metric
* Authors plan to release adversarial examples with annotations

Weaknesses:
* Since it mostly relies on LPIPS, it is unclear how much novelty there is in the approach in the NPTM.
* The PPGD and LPA attacks completely break the PAT (Perceptual Adversarial Training), so the reader is left with a bit of mixed feelings about lessons learned. I think the authors could have elaborated more on this.
* Fast-LPA seems to be a relevant contribution but is relegated to the Appendix.
* Related work is a bit succinct. Since the major proposal is about neural perceptual distance, but they rely on something from the state of the art, although extensively validation, it would be interesting to know more about what are the actual contributions with respect to this part as well. The new attacks against this NPTM are a clear contribution, in addition to validation with mechanical turk of the perceptual similarity.

Comments:
* As a general comment, I believe it would be clearer to report "attack success rate" instead of decreased accuracies, although I understand the work is a bit on attacks and a bit on the robustness side. So, to show relevance of adversarial training, eventual model accuracy is okay.
* It is a bit confusing that in Table 2 and 3, the “Normally” trained model has an Unseen Mean of 1.8% instead of 0%. I eventually understood why, but maybe you should consider commenting on this, as I initially felt like there was something wrong with the Table.
* The authors emphasize a lot on the NPTM, which is definitely relevant. Nevertheless, the PAT (Perceptual Adversarial Training) seems mostly unsuccessful against the PPGD and LPA (and Fast-LPA) attacks, and this may
* I am a bit confused by the overall narrative of the work. You first propose the NPTM, and develop the PAT mechanism, and show it generalizes well across different threat models. But then you develop an attack that “knows” the defense - which is great - but then show that it basically defeats PAT (mostly). I think this part of the story is left a bit under-explored, and the reader is missing actionable points from the work, as there is a lot of discussion on robustness but also on attacking this new perceptual models. Do not get me wrong: I think this is highly relevant, I am mostly commenting on the presentation of the results and the overall story and lessons learned.

---

> ### Author Response · Authors · 2020-11-18
> **Responses to your comments**
>
> Thank you for your insightful review. Here are responses to your comments:
>  * You note that "since it mostly relies on LPIPS, it is unclear how much novelty there is in the approach in the NPTM." Although LPIPS was previously proposed, it has mostly been used for development and evaluation of generative models (e.g. [1] and [2]). To the best of our knowledge, we are the first to apply a more accurate perceptual distance to the problem of adversarial robustness. Adversarial robustness has largely focused on $L_2$ or $L_\infty$ attacks, which as we show are unable to generalize to a more diverse threat model. Our method, PAT, is the first we know of that can generalize to unforeseen threat models. We have updated and expanded our related work section to clarify this.
>
>    As you mentioned, our specific technical contributions include developing the two perceptual attacks, PPGD and LPA, which are based on the NPTM. These require novel strategies compared to $L_\infty$ or $L_2$ adversarial examples since the constraint on the attack itself is defined by a neural network. Furthermore, we develop Fast-LPA, which is nearly as fast as a typical adversarial training PGD attack despite the difficulties in the NPTM constrained optimization problem. We also show that LPIPS is a good perceptual measure for adversarial examples through our Mechanical Turk study; this is a bit surprising, since one might think that adversarial examples would fool the neural network used for the LPIPS calculation.
>  * It is true that PPGD and LPA still have a higher success rate against PAT-trained models than the narrow attacks. We believe this is explained by a similar reason to that for the success of PAT: since the NPTM encompasses a wider set of potential adversarial examples, it is easier to find a successful adversarial example that causes misclassification within that set. For example, $L_\infty$ attacks are more difficult to defend at $\epsilon = \frac{16}{255}$ than at $\epsilon = \frac{8}{255}$, because it is easier to find an adversarial example within a ball of twice the radius. Analogously, the reason PPGD and LPA reduce even PAT-trained classifiers to such low accuracy is that the NPTM encompasses more potential adversarial examples than the narrow threat models.
>
>    To show this effect, we evaluated classifiers trained with adversarial training and PAT against LPA at smaller bounds. The table below shows the accuracy of some classifiers on ImageNet-100 against LPA with these bounds; the rightmost column is the bound used in the paper. Note that PAT significantly increases robustness against LPA compared to adversarial training across these three bounds.
>
>    |Training|LPA ($\epsilon = 0.125$)$\quad$|LPA ($\epsilon = 0.25$)$\quad$|LPA ($\epsilon = 0.5$)$\quad$|
>    |--|--|--|--|
>    | Normal | 2.0 | 0.0 | 0.0 |
>    | AT $L_\infty$ | 3.9 | 0.0 | 0.0 |
>    | AT $L_2$ | 53.4 | 19.8 | 0.5 |
>    | PAT-self | **61.5** | 28.1 | **2.4** |
>    | PAT-AlexNet$\quad$ | 60.5 | **28.7** | 1.6 |
>
>    We pose finding classifiers with good robustness against our PPGD and LPA attacks with the higher bounds as an open problem to the adversarial robustness community. This problem is particularly relevant because we find that robustness against PPGD/LPA is a good proxy for robustness against a range of other threat models. We will clarify in the paper that our contributions are both (a) a method to defend against unforeseen narrow threat models, PAT, and (b) new perceptual attacks, PPGD and LPA, which can help the community better evaluate robustness against a broader threat model.
>  * We have expanded the description of Fast-LPA in the main text based on your feedback.
>  * In Tables 2 and 3, the unseen mean accuracy of the normally trained model was actually calculated incorrectly. It should have been 0.4% and 0.1% for CIFAR-10 and ImageNet-100, respectively. We fixed this typo. Thank you for pointing this out.
>  * We report robust accuracy instead of attack success rate because, like you mentioned, we are focusing on defenses as well as attacks. We also find that "attack success rate" sometimes refers to `1 - robust_accuracy` and sometimes to `clean_accuracy - robust_accuracy`; thus, the term can cause some confusion. However, thank you for your suggestion to clarify the paper.
>
> [1] Zhang et al. Multimodal Unsupervised Image-to-Image Translation, ECCV 2018.
>
> [2] Karras et al. A Style-Based Generator Architecture for Generative Adversarial Networks, CVPR 2019.

---

> > ### Comment · AnonReviewer4 · 2020-11-23
> > **Feedback to response**
> >
> > Dear authors,
> >
> > thank you very much for clarifying your contributions/novelty and experimental evaluation. I am satisfied with your answer, and I hope you will integrate your response within the main text of the paper. I think it would greatly improve clarity, especially with respect to novelty with respect to the state of the art.

---

### Official Review · AnonReviewer2 · 2020-10-29
**Good work**

**Rating:** 7
**Confidence:** 3

**Review:**

This work proposes a new form of adversarial training, supported by two proposed adversarial attacks based off a perceptual distance. The choice of perceptual distance (LPIPS), is computed by comparing the activations of (possibly different) two neural networks with respect to a pair of inputs. The authors propose two new attacks based off this perceptual distance: PPGD and LPA, as it is distinct from the common choice of L_2 or L_inf. This work claims that performing adversarial training against adversarial examples crafted by the proposed attacks, induces robustness to a wide range of "narrow" threat models e.g. L_2, JPEG, L_inf.
To show this, the authors perform experiments on CIFAR-10 and ImageNet-100 — with the main results being that adversarial training with PPGD or LPA produces a model with some robustness to all other threat models. This is in contrast to adversarial training with "narrow" threat models, which fail to be robust to at least one other threat model in the set. The exception possibly being L_2, which retains some transferability.

The paper is written well, and is somewhat easy to follow.

Overall I believe this to be a good paper. I appreciate the study of a union of threat models, as in practice, we of course have no guarantee that an adversary will choose to restrict itself to a single threat model. The experimental results are compelling, and match my intuition — namely that adversarial training with the L_2 threat model would produce the most comparable results to a perceptual threat model, when comparing against the union. Moreover, I find it very interesting that while L_2 looks to transfer well to other threat models, it does not transfer comparably well to the perceptual threat model. The authors explain that this is due to the "broadness" of the threat model, but this could use some further exposition.

Qualitatively, the LPA and PPGD adversarial examples appear distinct from other threat models in terms of the modified features. The LPA adversarial examples in particular to not seem to target local texture patches, which is nice to see.

I also appreciate the addition of less computationally intensive algorithms to support the proposed attacks. PPGD in particular would be very difficult to run, as the jacobian of \phi(x) can be very large.

I have some questions, in order of importance.

1) Why is AlexNet (externally-bounded) more effective than using the same network to compute LPIPS? This may be because the metric changes as the network trains, but then I would expect the self-bounded training to achieve worse clean accuracy if the metric degraded.

2) What is the reason for believing that the neural perceptual threat model encompasses the L_p and spatial threat models? I see that there is some overlap in terms of how each threat model can transfer to other threat models. But it is not obvious that the "narrow" threat models are contained with the perceptual threat model.

3) I think its interesting that PAT-* performs worse on PPGD and LPA, than on other threat models (that presumably the PAT-* model has not seen). This could mean that PPGD and LPA adversarial examples are "harder" in some way. Do the authors have more explanation/intuition about this?

More general comments:

I found parts of the paper somewhat difficult to read due to having to check the appendix often. It would be easier to read if some details like the attack algorithms, and a figure showing the attack process were moved to the main paper.

---

> ### Author Response · Authors · 2020-11-18
> **Responses to your questions**
>
> Thank you for your insightful review. Here are responses to your questions:
>
>  1. As you observed, performing externally-bounded PAT with AlexNet produces more robust models on CIFAR-10 than self-bounded PAT. This is not the case on ImageNet-100, where self- and AlexNet-bounded PAT perform similarly.
>
>     There is a simple explanation for this: the effective training bound on CIFAR-10 is greater for AlexNet-bounded PAT than for self-bounded PAT. To measure this, we generate adversarial examples for all the test threat models on CIFAR-10 ($L_2$, $L_\infty$, StAdv, and ReColorAdv). We find that the average LPIPS distance for all adversarial examples using AlexNet is 1.13; for a PAT-trained ResNet-50, it is 0.88. Because of this disparity, we use a lower training bound for self-bounded PAT (0.5) than for AlexNet-bounded PAT (1). However, this means that the average test attack has 76% greater LPIPS distance than the training attacks for self-bounded PAT, whereas the average test attack has only 13% greater LPIPS distance for AlexNet-bounded PAT. This explains why AlexNet-bounded PAT gives better robustness; it only has to generalize to slightly larger attacks on average.
>
>    We tried performing AlexNet-bounded PAT with a more comparable bound (0.7) to self-bounded PAT. This gives the average test attack about 80% greater LPIPS distance than the training attacks, similar to self-bounded PAT. As you can see below, the results are now more similar for self-bounded and AlexNet-bounded PAT. We have included this discussion along with the evaluation of PAT using different bounds in Appendix F.3.
>
>    |Training method|Union$\quad$|Unseen Mean$\quad$|Clean$\quad$|$L_\infty$$\quad$|$L_2$$\quad$|StAdv$\quad$|ReColorAdv$\quad$|PPGD$\quad$|LPA$\quad$
>    |--|--|--|--|--|--|--|--|--|--|
>    |PAT-self ($\epsilon = 0.5$)|21.9|45.6|82.4|30.2|34.9|46.4|71.0|13.1|2.1|
>    |PAT-AlexNet ($\epsilon = 0.7$)|16.7|45.3|80.0|35.5|41.3|33.2|71.5|17.8|4.9|
>    |PAT-AlexNet ($\epsilon = 1.0$)$\quad$|27.8|48.5|71.6|28.7|33.3|64.5|67.5|26.6|9.8|
>
>   2. You ask, "what is the reason for believing that the neural perceptual threat model encompasses the L_p and spatial threat models?" In a new figure (Figure 5, see the updated paper), we show how the narrow threat models are nearly contained within the NPTM. As can be seen in Figure 5(b), most adversarial examples from other threat models are far in $L_2$ distance from natural inputs. In order to be robust against such threat models, $L_2$ adversarial training would have to use a very large training bound. In contrast, adversarial examples from all threat models have similar LPIPS distance from natural inputs (Figure 5(c-d)). Thus, PAT can give robustness against all threat models at a single training bound. This explains why PAT produces greater robustness against these unseen threat models.
>   3. It is true that PPGD and LPA still have a higher success rate against PAT-trained models than the narrow attacks. We believe this is explained by a similar reason to that for the success of PAT: since the NPTM encompasses a wider set of potential adversarial examples, it is easier to find a successful adversarial example that causes misclassification within that set. For example, $L_\infty$ attacks are more difficult to defend at $\epsilon = \frac{16}{255}$ than at $\epsilon = \frac{8}{255}$, because it is easier to find an adversarial example within a ball of twice the radius. Analogously, the reason PPGD and LPA reduce even PAT-trained classifiers to such low accuracy is that the NPTM encompasses more potential adversarial examples than the narrow threat models.
>
>    To show this effect, we evaluated classifiers trained with adversarial training and PAT against LPA at smaller bounds. The table below shows the accuracy of some classifiers on ImageNet-100 against LPA with these bounds; the rightmost column is the bound used in the paper. Note that PAT significantly increases robustness against LPA compared to adversarial training across these three bounds.
>
>    |Training|LPA ($\epsilon = 0.125$)$\quad$|LPA ($\epsilon = 0.25$)$\quad$|LPA ($\epsilon = 0.5$)$\quad$|
>    |--|--|--|--|
>    | Normal | 2.0 | 0.0 | 0.0 |
>    | AT $L_\infty$ | 3.9 | 0.0 | 0.0 |
>    | AT $L_2$ | 53.4 | 19.8 | 0.5 |
>    | PAT-self | **61.5** | 28.1 | **2.4** |
>    | PAT-AlexNet$\quad$ | 60.5 | **28.7** | 1.6 |
>
>    We pose finding classifiers with good robustness against our perceptual attacks (PPGD and LPA) with the higher bounds as an open problem to the adversarial robustness community. This problem is particularly relevant because we find that robustness against PPGD/LPA is a good proxy for robustness against a range of narrow threat models. We will clarify in the paper that our contributions are both (a) a method to defend against unforeseen narrow threat models, PAT, and (b) new perceptual attacks, PPGD and LPA, which can help the community better evaluate robustness against a broader threat model.

---

### Public Comment · ~Zhengyu_Zhao1 · 2020-11-16
**Adaptive attacks**

Thanks for this interesting work.

Concern:

If I understand correctly, this paper claims white-box robustness, i.e, their Perceptual Adversary-based defense is potentially effective against other white-box attacks that are also based on gradient optimization.

However, the authors completely ignore the evaluation against adaptive attacks, which is NECESSARY to claim justified white-box robustness as suggested by the best practices in the literature [1][2][3].
For example, [2] states that "Defending against non-adaptive attacks is necessary but not sufficient. It is our firm belief
that an evaluation against non-adaptive attacks is of very limited utility."
In other words, if the attacker knows the strategy used by the defender, it may be possible to break the model. And so, robustness against several existing white-box attacks can not yield generalizable effectiveness against future white-box attacks.

I would appreciate it if the authors could provide experimental results on the adaptive attacks, which are specifically designed to break the proposed defense algorithm.

[1] Athalye, Anish, Nicholas Carlini, and David Wagner. "Obfuscated gradients give a false sense of security: Circumventing defenses to adversarial examples." ICML 2018.
[2] Carlini, Nicholas, et al. "On evaluating adversarial robustness." arXiv 2019.
[3] Tramer, Florian, et al. "On adaptive attacks to adversarial example defenses." NeurIPS 2020.

---

> ### Author Response · Authors · 2020-11-18
> **Our evaluation is already exhaustive**
>
> Thank you for your comment. We agree that evaluating adversarial defenses against adaptive attacks is very important. We believe we are already performing an exhaustive evaluation of our defense for a few reasons.
>
> First, we test against two novel perceptual attacks designed specifically for our neural perceptual threat model (NPTM), LPA and PPGD. These adaptive attacks are in fact the strongest against our PAT defense, indicating that we are using a strong evaluation attack.
>
> Second, our defense is not based on changes to the classifier architecture or inference procedures. We train standard ResNet-50s and use them normally for inference. The only difference in our defense (PAT) compared to adversarial training is the training attack used. Thus, we believe that the strongest attacks against adversarial training-type defenses, which are PGD-based, should also suffice to be a good evaluation of our defense.
>
> Finally, we use AutoAttack [1] to test robustness against $L_\infty$ and $L_2$ adversarial attacks. We use the authors' original implementation, which performs four strong gradient-based and gradient-free attacks against every input. It has been shown to be the strongest attack against a number of adversarial defenses, so we believe using it to evaluate PAT gives a fair measure of robustness.
>
> Please let us know if you have more specific concerns about any aspects of our evaluation procedures.
>
> [1] Croce and Hein. Reliable evaluation of adversarial robustness with an ensemble of diverse parameter-free attacks. ICML, 2020.

---

### Comment · ~Martin_Rinard1 · 2021-05-14
**Previous Work On This Topic**

The last sentence of the abstract states: "PAT is the first adversarial defense with this property" where "this property" is apparently "generalizes well to unforeseen perturbation types." The paper https://arxiv.org/abs/2003.04286, with a publication date of March 2020, uses a different approach (nonadversarial training), but does present a technique that generalizes to unforeseen perturbation types under the l_\infty, l_2, and Wasserstein metrics.

---

> ### Author Response · Authors · 2021-05-17
> **Will clarify the claim**
>
> We were unaware of your paper when we first wrote ours, but we can revise the paper to clarify that ours is the "first adversarial *training* defense with this property". We will also cite your paper in the related work. Would it be possible for you to share your pretrained models? It would be great to include a comparison to your work in the results.

---

> > ### Comment · ~Charles_Jin1 · 2021-06-05
> > **Pretrained models**
> >
> > Hi all,
> >
> > Thanks for the response and sorry about the delay. I've uploaded our model and weights to dropbox below.
> >
> > https://www.dropbox.com/sh/jfqo2l1i2dyj3o9/AABWq8IQZmQlMLJ_ntMj-TYVa?dl=0
> >
> > FYI, here are the results we get using the attack methods in the PAT paper:
> >
> > |OVERALL|
> > | --- |
> > |ATTACK NoAttack()	 accuracy = 72.1|
> > |ATTACK AutoLinfAttack(model, 'cifar', bound=8/255)	 accuracy = 36.8|
> > |ATTACK AutoL2Attack(model, 'cifar', bound=1)	 accuracy = 43.4|
> > |ATTACK StAdvAttack(model, num_iterations=100)	accuracy = 28.4|
> > |ATTACK ReColorAdvAttack(model, num_iterations=100)	accuracy = 63.1|
> > |ATTACK PerceptualPGDAttack(model, num_iterations=40, bound=0.5, lpips_model='alexnet_cifar', projection='newtons')	accuracy = 8.5|
> > |ATTACK LagrangePerceptualAttack(model, num_iterations=40, bound=0.5, lpips_model='alexnet_cifar', projection='newtons')	accuracy = 9.1|
> >
> > Please let us know if your results are materially different or you're having issues getting our model working. And a huge thank you for making it so easy to reproduce your results

---

> > > ### Author Response · Authors · 2021-07-05
> > > **Paper has been updated**
> > >
> > > Hello,
> > >
> > > Thank you for providing your model and evaluation. We reproduced your results and have included them in an updated version of the paper, along with other edits to clarify the claims in the abstract and include your paper in the related work. The new version is uploaded here on OpenReview and will also be updated on arXiv later today.

---

### Decision · Program_Chairs · 2021-01-07
**Final Decision**

**Decision:**

Accept (Poster)

**Comment:**

This paper focuses on the adversarial robustness of deep neural networks against multiple and unforeseen threat models, which proposes a threat model called Neural Perceptual Threat Model (NPTM). The philosophy behind sounds quite interesting to me, namely, approximating human perception with a neural neural "neural perceptual distance". This philosophy leads to a novel algorithm design I have never seen, i.e., Perceptual Adversarial Training (PAT) which achieves good robustness against various types of adversarial attacks and even could generalize well to unforeseen perturbation types.

The clarity and novelty are clearly above the bar of ICLR. While the reviewers had some concerns on the significance, the authors did a particularly good job in their rebuttal. Thus, all of us have agreed to accept this paper for publication! Please carefully address all
comments in the final version.